# No Labels, No Problem: Training Visual Reasoners with Multimodal Verifiers

**Damiano Marsili, Georgia Gkixoari**
California Institute of Technology

## Abstract

Visual reasoning is challenging, requiring both precise object grounding and understanding complex spatial relationships. Existing methods fall into two camps: language-only chain-of-thought approaches, which demand large-scale (image, query, answer) supervision, and program-synthesis approaches which use pretrained models and avoid training, but suffer from flawed logic and erroneous grounding. We propose an annotation-free training framework that improves both reasoning and grounding. Our framework uses AI-powered verifiers: an LLM verifier refines LLM reasoning via reinforcement learning, while a VLM verifier strengthens visual grounding through automated hard-negative mining, eliminating the need for ground truth labels. This design combines the strengths of modern AI systems: advanced language-only reasoning models for decomposing spatial queries into simpler subtasks, and strong vision specialist models improved via performant VLM critics. We evaluate our approach across diverse spatial reasoning tasks, and show that our method improves visual reasoning and surpasses open-source and proprietary models, while with our improved visual grounding model we further outperform recent text-only visual reasoning methods. Project webpage: https://glab-caltech.github.io/valor/

## 1 Introduction

Visual reasoning is a key skill for artificial intelligence: to understand and act in the world, systems must not only identify objects in images but also reason about their spatial relationships and attributes. For example, answering the query in Fig. 1 requires grounding objects (fireplace, coffee table, sofa), inferring 3D size from 2D cues, and combining attributes to produce the final answer.

Visual reasoning methods fall into two categories. The first integrates grounding with language reasoning, where vision-language models (VLMs) generate chain-of-thought explanations in text (Fan et al., 2025; Sarch et al., 2025; OpenAI, 2025b). These methods can handle simple spatial relations, but suffer from weak visual understanding and logical errors. For example, in Fig. 1, GPT-5-Thinking ignores real-world 3D object sizes and considers only pixel-wise dimensions, incorrectly concluding the coffee table is six times shorter than the sofa. These methods are also data-hungry, requiring extensive supervision. Another line of work uses LLMs for program synthesis with vision specialists (Surís et al., 2023; Gupta & Kembhavi, 2023; Marsili et al., 2025), but relies on proprietary LLMs and pre-trained specialists that are poorly aligned with spatial reasoning.

In this work, we tackle visual reasoning via tool use through a *scalable, annotation-free* training framework that jointly tunes reasoning LLMs and vision tools for spatial understanding. Drawing inspiration from reinforcement learning with verifiable rewards for mathematical reasoning (Jaech et al., 2024; Guo et al., 2025; Team et al., 2025), we design verifiers to form structured reward models which capture correct logical decomposition, tool use and syntax. We show this reward model acts as a strong learning guide for spatial reasoning in the absence of ground truth labels. We name our approach VALOR as it integrates Verifiers for Annotation-free LOgic and Reasoning. As Fig. 1 shows, VALOR accurately invokes visual grounding tools, converts 2D to 3D measurements by integrating object depth, and accurately combines the measurements to produce the right answer.

Accurate visual grounding is critical for spatial reasoning, as localization errors propagate through later steps. Vision specialists are pre-trained on web data that diverge from spatial reasoning domains like robotics and manipulable objects. Fine-tuning on target domains is onerous due to annotation cost. To address this, we extend the verifier-based principle to the task of visual grounding. A

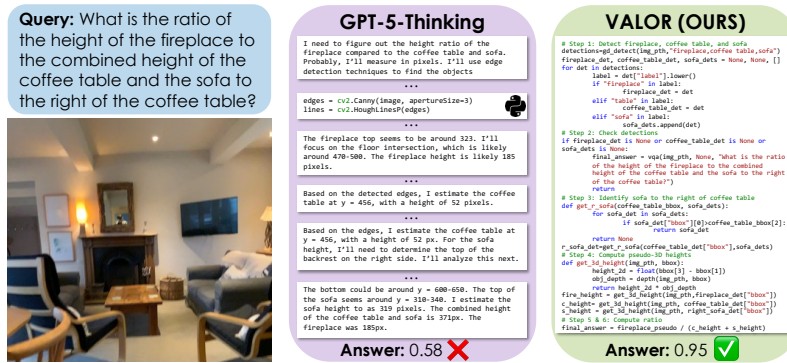

Figure 1: Visual reasoning relies on accurate reasoning and visual grounding. To tackle the task, we propose an annotation-free training paradigm, called VALOR, that learns to decompose the task and invoke tools by leveraging multimodal verifiers, without the need of ground truth supervision.

VLM-based verifier refines the outputs of a grounding model and refined predictions are recycled as training data. This loop strengthens grounding for spatial reasoning, building on ideas from bootstrapping and hard-negative mining in face detection (Sung, 1996; Sung & Poggio, 1998) and object recognition (Rosenberg et al., 2005; Shrivastava et al., 2016; Radosavovic et al., 2018).

We evaluate VALOR on a broad range of spatial reasoning benchmarks with natural images and show that our verifier-powered annotation-free training framework improves reasoning when compared to both open-source LLMs and larger proprietary models. Additionally, we show that through reasoning and tool use, VALOR outperforms VLMs that either predict answers directly or are specifically tuned for visual grounding. Through a scaling analysis, we show that verifier-powered training signal has an upward trend in both reasoning and execution accuracy, offering a scalable alternative to collecting expensive ground truth labels for visual tasks.

## 2 RELATED WORK

**Tool Use in VLMs.** While VLMs (Anthropic, 2024; OpenAI, 2025a; Team et al., 2023) demonstrate strong performance on visual captioning, they struggle with complex spatial reasoning tasks (Rahmanzadehgervi et al., 2024; Ray et al., 2024; Thrush et al., 2022; Wu & Xie, 2024; Tong et al., 2024). Visual programming approaches tackle spatial reasoning by generating interpretable, executable programs that invoke vision specialists (Gupta & Kembhavi, 2023; Surís et al., 2023; Chen et al., 2024; Marsili et al., 2025), but suffer from faulty program logic and poor execution as they rely on pre-trained models not well-suited for the task. VALOR also generates programs and invokes vision specialist tools, but aligns models to the task by tuning them with multimodal verifiers.

**Reinforcement Learning for Reasoning.** Reinforcement learning (RL) has enhanced LLM reasoning in domains with verifiable rewards, such as mathematics and programming (Jaech et al., 2024; Guo et al., 2025; Team et al., 2025), using tailored algorithms (Yu et al., 2025; Liu et al., 2025b). While RL for text-based reasoning is well studied (Wang et al., 2024a; Cui et al., 2025; Lambert et al., 2024), visual reasoning remains underexplored. Recent works extend RL to visual tasks (Sarch et al., 2025; Fan et al., 2025; Dong et al., 2025; OpenAI, 2025b) by interleaving visual grounding with text-based reasoning. However, our experiments show these methods struggle with grounding and reasoning, while requiring ground truth labels to train. We show VALOR outperforms them in both reasoning and grounding, with the largest gains in 3D spatial understanding.

**Model Distillation.** Methods in knowledge distillation typically trains a student model to imitate a teacher's output distributions (Zelikman et al., 2022; Li et al., 2024; Muennighoff et al., 2025) or internal representations (Hu et al., 2024; Xu et al., 2025; Sarch et al., 2024), offering dense target-level supervision. In contrast, our work uses pretrained models as verifiers that provide feedback over intermediate outputs, providing reinforcement-based optimization rather than teacher imitation.

**Training with LLM Verifiers.** LLM verifiers provide pseudo-supervision when ground truth labels are unavailable, with self-verification methods SCoRe (Zweiger et al., 2025), SQLM (Chen et al.,

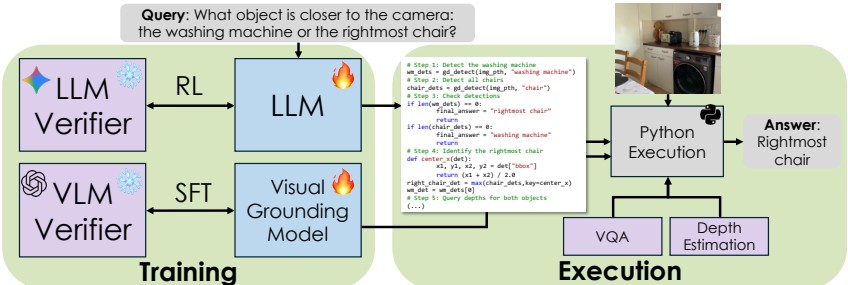

Figure 2: **Method overview.** Our method, VALOR, tackles visual reasoning across a broad range of spatial tasks, in 2D and 3D. During training, LLM verifiers are used to improve reasoning via RL while VLM verifiers serve as critics to tune vision grounding models via SFT.

2025b), and RISE (Qu et al., 2024) generating synthetic data for RL training. SEAL (Zweiger et al., 2025) applies this idea to supervised fine-tuning. SWiRL (Goldie et al., 2025) extends this approach by using high-capacity verifiers to improve low-capacity models through multi-step RL, while other methods (Liu et al., 2025a; Chen et al., 2025a; Zhao et al., 2025) leverage verifiers at inference time.

**Training with VLM verifiers.** Recent work integrates VLMs into vision model training to guide training. SemiVL (Hoyer et al., 2024) injects VLM guidance to semantic segmentation. In object detection, PB-OVD (Gao et al., 2022) uses VLMs to generate pseudo-labels from image-caption pairs. Others propose filtering: VLM-PL (Kim et al., 2024) and AIDE (Liang et al., 2024) use VLMs to curate candidate labels for training, and MarvelOVD (Wang et al., 2024b) uses the detector to verify VLM outputs. While prior work uses VLM-guided refinement, our approach unifies reasoning and vision tuning via LLM and VLM verifiers, respectively, with a focus on spatial reasoning.

## 3 TRAINING VISUAL REASONERS WITH VERIFIERS

We tackle spatial reasoning from images by combining LLM-powered reasoning with specialized tool use. VALOR employs an LLM to generate plans and executable programs and invokes vision specialists for execution. Both the reasoning and the vision grounding model are tuned for the task via a label-free training paradigm. This is achieved by leveraging multimodal verifiers that critique model outputs. Their feedback serves as a learning signal to improve both components, the LLM responsible for logic and the vision specialists responsible for grounding. Fig. 2 shows our approach.

**Plan & Code Generation.** Given a query, the LLM generates a natural language plan followed by a corresponding program in Python. Available to the LLM are the APIs of three function calls:
- GD_DETECT, returns the bounding box of all object instances specified by the noun description – e.g., GD_DETECT("CAR"),
- DEPTH, returns the depth of a pixel in the image – DEPTH(IMAGE, X, Y)
- VQA, returns an object's attribute (e.g., color) from the input image crop around the object – e.g., VQA(IMAGE_CROP, "WHAT IS THE COLOR OF THE OBJECT IN THE IMAGE?")

The API specifications are provided in the prompt to the LLM (see Appendix A.10). Plans and programs are delimited by `<plan></plan>` and `<answer></answer>` tags, respectively.

**Vision Specialists.** VALOR employs three vision specialist models: GroundingDINO (Liu et al., 2023b) for object localization (GD_DETECT), MoGe2 (Wang et al., 2025) for pointwise metric depth estimation (DEPTH), and GPT-5-mini (OpenAI, 2025a) for VQA with an image cropped around an object bounding box (VQA). Implementation details are in Appendix A.2.

### 3.1 IMPROVING REASONING WITH LLM VERIFIERS

Visual reasoning requires decomposing tasks into subtasks, correctly invoking APIs, and applying perceptual principles (e.g., distant objects appear smaller). However, state-of-the-art LLMs struggle across all three axes: they fail to break down queries into coherent steps, misuse tools, and neglect basic perceptual principles. Fig. 3(a) shows program generated by a pre-trained LLM, Qwen3-8B (Yang et al., 2025): it misses the spatial object relationship (shelf "below" sink) and considers

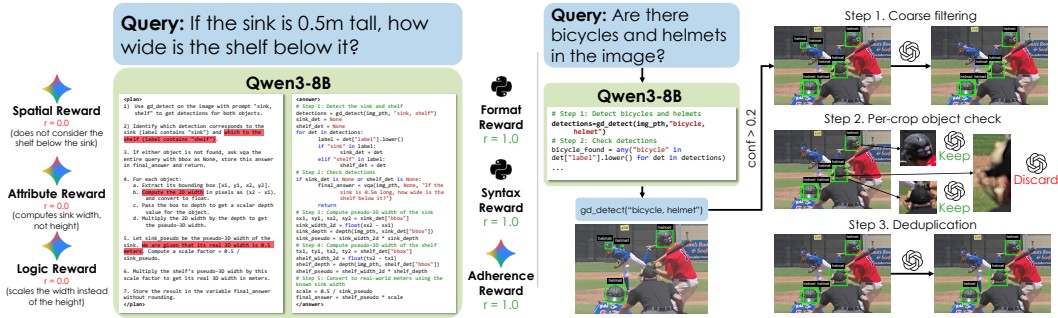

(a) LLM verifier                  (b) VLM verifier

Figure 3: **(a)** LLM verifiers reward semantic correctness by evaluating logical correctness, object attribute and spatial relationship consideration, and if the code adheres to the predicted plan. Python interpreters check format adherence and syntax. **(b)** VLM verifiers refine visual grounding over-predictions through three stages, generating pseudo-labels from spatial reasoning queries.

wrong object dimensions (sink width instead of height). To refine reasoning, we introduce verifiers that provide feedback on the generated programs. We design a reward model specialized for spatial reasoning and use it to fine-tune the LLM to improve both program decomposition and tool usage.

**Setup.** Given a query $q$, our base LLM $\pi_\theta$, parametrized by $\theta$, generates a natural language plan $p$ and corresponding Python code $c$ with access to predefined tools via their API $\mathcal{A}$: $(p, c) = \pi_\theta(q; \mathcal{A})$. We aim to find a model, $\pi_{\theta^*}$, that optimizes program correctness, API usage and coordination.

**Reward Model.** We design a reward model which *verifies* whether model outputs – plan & code – are correct. Our reward decomposes program quality into six binary components, each targeting specific aspects of spatial reasoning: (1) **Format** $r_{\text{fmt}}(p, c) = 1$ if output follows `<plan></plan><answer></answer>` template, 0 otherwise; (2) **Syntax** $r_{\text{sn}}(c) = 1$ if code executes without Python errors, 0 otherwise; (3) **Logic** $r_{\text{log}}(q, p) = 1$ if verifier considers the plan reasonable and coherent for query $q$, 0 otherwise; (4) **Attribute** $r_{\text{att}}(q, p) = 1$ if all object properties (height, color, etc.) are correctly identified from the query, 0 otherwise; (5) **Spatial** $r_{\text{sp}}(q, p) = 1$ if plan addresses all spatial relationships in query, 0 otherwise; and (6) **Adherence** $r_{\text{ad}}(p, c) = 1$ if the code faithfully implements the plan without deviations, 0 otherwise. The final reward is:

$$R(q, p, c) = r_{\text{fmt}}(p, c) \cdot \left[ \lambda_{\text{sn}} r_{\text{sn}}(c) + \lambda_{\text{log}} r_{\text{log}}(q, p) + \lambda_{\text{att}} r_{\text{att}}(q, p) + \lambda_{\text{sp}} r_{\text{sp}}(q, p) + \lambda_{\text{ad}} r_{\text{ad}}(p, c) \right]$$
(1)

The format reward $r_{\text{fmt}}$ acts as a hard constraint and is applied as a multiplier, while the weighted sum of the remaining rewards evaluates content quality. All $r_k \in \{0, 1\}$ and $\sum_k \lambda_k = 1.0$.

Format and syntax rewards use deterministic python interpreters (where tool calls are replaced with dummy functions), while reward components (3)-(6) require semantic evaluation, and thus employ a frozen, pre-trained LLM as the verifier model with task-specific prompts that output binary decisions. Fig. 3(a) shows the output of our reward model: the verifier highlights the errors – in spatial, attribute and logic decomposition – and the correct components – in format, syntax and adherence. We highlight the role of each reward head and provide further details in Appendix A.6.2.

**Optimization.** We optimize our LLM $\pi_\theta$ using GRPO (Guo et al., 2025). GRPO maximizes expected advantages while maintaining policy stability by preventing drift from a pre-trained base policy. We provide further details in Appendix A.6.1.

**Generating training data.** We train on image–query pairs, $\{(I_j, q_j)\}_{j=1}^N$, without requiring ground-truth answers, using the reward model as the sole learning signal. This design enables training to extend beyond the small labeled datasets available for spatial reasoning to arbitrary image corpora. We sample real-world images from SA-1B (Kirillov et al., 2023) and prompt Gemini-2.5-Flash (Comanici et al., 2025) to generate five queries per image, and uniformly select one. This process pairs unlabeled images to spatial reasoning queries. An example generated query includes *"How many animals are visibly seated within the beige vehicle?"*. More details and examples of generated queries are in Appendix A.4. To ensure strong coverage of 3D spatial reasoning, we additionally include (image, query) samples from OMNI3D-BENCH (Brazil et al., 2023), omitting the answers.

In practice, we found training on a few hundred queries is optimal for GRPO. Our final dataset contains 800 (image, query) pairs: 400 generated from SA-1B via our query generation engine and 400 from OMNI3D-BENCH. We show the impact of varying training data in our experiments.

**Error Analysis.** As verifiers provide the learning signal in VALOR, it is essential to assess their reliability. Thus, we manually annotate rewards for 100 model outputs and treat these as ground truth. Gemini-2.5-Flash – the LLM verifier in VALOR – agrees with these annotations in $87\%$ of cases, with most mismatches due to Gemini under-rewarding. Two open-source verifiers perform worse, matching ground truth only $15\%$ (Qwen3-8B) and $7\%$ (Llama-3.2-11B-Instruct) of the time. This gap highlights the need for high-capacity verifiers and supports our choice of Gemini-2.5-Flash.

**Implementation.** We use Qwen3-8B as our base LLM $\pi_\theta$, pre-trained for language-only tasks (Yang et al., 2025). We train on 8 A100 GPUs for 4 epochs with a batch size of 64, group size $G = 5$, and learning rate of $10^{-6}$. We use verl (Sheng et al., 2025) for training. All hyper-parameters are in Appendix A.6. We denote this trained model VALOR-RL in all experiments.

## 3.2 IMPROVING VISUAL GROUNDING WITH VLM VERIFIERS

In addition to logic, visual reasoning relies on accurate grounding. While math reasoning relies on error-free tools like calculators, spatial reasoning depends on imperfect visual grounding models such as object detectors. A major failure mode, also observed in Marsili et al. (2025), is incorrect grounding of target objects. This is not surprising as visual grounding is present in all queries, from counting to complex 3D reasoning. Modern detectors like GroundingDINO (Liu et al., 2023b), trained on web data, are error-prone and struggle to generalize beyond their training domains. Fig. 3(b) shows GroundingDINO's output for "bicycle" and "helmet" from the input query. While it correctly finds no bicycles, it misclassifies baseball caps as helmets. Fine-tuning with domain-specific labels can mitigate these issues, but collecting such annotations is labor intensive.

We propose an alternative: improving visual grounding through VLM verifiers. Vision specialists cast predictions, VLM verifiers evaluate them, and the feedback augments their training set. This approach requires no manual annotations and scales across domains without additional labels.

**VLM verifiers as data labelers.** We use VLM verifiers to generate pseudo-annotations for object detection that better capture the object distribution and target domain for spatial reasoning. We rely on image–query pairs $\{(I_j, q_j)\}_{j=1}^{M}$. For each query $q_j$, our LLM reasoning model generates a plan and code, $(p_j, c_j)$. From code $c_j$, we parse all grounding queries – e.g., GD_DETECT("HELMET") – and execute them with a pre-trained detector. To ensure high recall, we lower the detector's confidence threshold. This leads to overprediction, which we validate with a frozen VLM in three steps: (1) *coarse filtering* removes invalid detections given the image and boxes overlaid, (2) *per-crop verification* validates remaining detections on cropped regions, and (3) *de-duplication* eliminates duplicates. Confirmed detections form the new training set. Per-stage outputs are shown in Fig. 3(b). See more examples in Fig. 11 in Appendix A.4.2.

**Generating training data.** Our approach requires no ground truth object annotations. Instead we use spatial reasoning queries – without answers – to gather detection signals through VLM verifiers. Since few images come paired with reasoning queries, we scale training through (image, query) pairs generated as in §3.1. We generate thousands of (image, query) pairs, which result in thousands of object annotations, verified by VLMs. Leveraging VLM verifiers as pseudo-labelers allows us to scale object detection training annotation-free, going beyond specialized spatial reasoning datasets to large, unconstrained image collections.

We augment the generated (image, query) pairs with pairs from small, existing training sets of spatial reasoning datasets, to improve generality and coverage of pseudo-labels. In total, our training set consists of 7,373 images and yields a total of 30,826 bounding box annotations spanning a broad range of object categories. Examples of the generated data are provided in Appendix A.4.

**Error Analysis.** We assess our three-stage VLM verification pipeline by measuring label precision on 100 random pipeline samples. For each stage, we compute precision from manually identified true positives (TP), false positives (FP), and false negatives (FN) using $P = \frac{TP}{TP+FP+FN}$. Precision increases at every stage: $0.45$ after coarse filtering, $0.50$ after per-crop verification, and $0.75$ after

| | OMNI3D-BENCH | ROBOSPATIAL | BLINK | VSR | REALWORLD QA | GQA | TALLY QA | COUNTBENCH QA |
|---|---|---|---|---|---|---|---|---|
| *Proprietary Models* | | | | | | | | |
| GPT-4o | 38.0 | 56.6 | 64.2 | 67.4 | 54.5 | 58.0 | 49.9 | **67.6** |
| o4-mini | 35.6 | 61.8 | **66.5** | **68.5** | **65.9** | 65.0 | 47.9 | 66.8 |
| Gemini-2.0-Flash | 37.0 | 57.0 | 51.7 | 58.5 | 47.3 | 52.1 | 43.6 | 62.1 |
| Gemini-2.5-Flash | 37.1 | **68.7** | 61.5 | **68.5** | 62.2 | **65.2** | 48.9 | 65.6 |
| Claude-3.5-Haiku | **43.6** | 55.7 | 64.6 | 51.5 | 54.4 | 61.3 | **50.1** | 65.2 |
| *Open-Source Models* | | | | | | | | |
| Llama-3.2-11B | 32.0 | 58.3 | 54.3 | 58.5 | 46.9 | 39.9 | 47.7 | 66.2 |
| Gemma-3-12B | 24.4 | 54.0 | 57.4 | 57.9 | 47.3 | 46.0 | 48.9 | 67.8 |
| Qwen3-8B | 37.5 | 60.5 | 63.9 | 68.2 | 53.3 | 57.4 | 50.1 | 68.6 |
| **VALOR-RL (ours)** | 43.9 | 61.8 | 67.3 | 70.3 | 53.5 | 57.6 | 49.5 | 67.6 |
| **VALOR (ours)** | **44.0** | **69.5** | **69.2** | **75.6** | **57.3** | **64.4** | **51.0** | **75.9** |

Table 1: **VALOR vs LLMs with tool use.** Language-only models generate Python programs with access to the same API of vision specialist models as VALOR. We highlight the **best** and second best proprietary model and the **best** and second best open-source model.

de-duplication. Our final stage of de-duplication yields the largest improvements ($+0.25$) – this is by design, our pipeline hinges on the detector over-predicting boxes, which are subsequently pruned by the verifier. The pipeline achieves a final precision of $75\%$ without any human annotations.

**Implementation.** We use our generated data to fine-tune GroundingDINO (Liu et al., 2023b). We use the `GroundingDINO-T` variant with a Swin Transformer (Liu et al., 2021) vision backbone and BERT (Devlin et al., 2019) language encoder. We keep the vision and language encoder frozen and train the rest of the model on 4 A100 GPUs for 7 epochs with a batch size of 16 and learning rate of $10^{-6}$. Details are in Appendix A.6. We show the impact of training data size in our experiments.

# 4 EXPERIMENTS

We experiment on a wide range of visual reasoning tasks that test VALOR's reasoning and grounding capabilities. Our goals are threefold: (1) to measure if verifier-driven RL improves task decomposition and tool use over strong LLM baselines; (2) to assess if multimodal verifiers improve visual grounding without annotations; and (3) to compare our approach against state-of-the-art alternatives, including supervised RL-tuned VLMs, program synthesis methods, and direct-answer models.

**Evaluation Benchmarks.** We evaluate VALOR on diverse real-world visual reasoning tasks covering different aspects of spatial understanding: OMNI3D-BENCH (Marsili et al., 2025) for 3D spatial understanding; TALLYQA (Acharya et al., 2019) and COUNTBENCHQA (Beyer et al., 2024) for counting; GQA (Hudson & Manning, 2019), REALWORLDQA (xAI, 2024), and VSR (Liu et al., 2023a) for 2D spatial relationships and image understanding. We include relevant single-image tasks from ROBOSPATIAL (Song et al., 2025) (compatibility, configuration) and BLINK (Fu et al., 2024) (counting, spatial relationships). For OMNI3D-BENCH, we reserve 100 queries for zero-shot evaluation and use the remaining 400 queries, without answers, to train our LLM with RL (§3.1). We report exact-match accuracy for yes/no, multiple choice, and integer response queries, and MRA for floating point response queries as in Marsili et al. (2025). Additional details are in Appendix A.3.

**Baselines.** We evaluate against four families of methods:

(1) LLMs with tools: We prompt proprietary and open-source LLMs (language-only) with a query to generate plans and programs using our vision-specialist APIs. LLMs output natural language plans in `<plan>` tags and programs in `<answer>` tags. We execute these programs and measure accuracy. This comparison highlights the gain of augmenting LLM reasoning with verifiers.
(2) Visually grounded, RL-tuned VLMs: GRIT (Fan et al., 2025) and ViGoRL (Sarch et al., 2025) finetune VLMs for visual grounding tasks but rely on ground truth labels for training. This comparison evaluates our label-free approach against RL-tuned methods with labels.
(3) Visual program synthesis methods: ViperGPT (Surís et al., 2023), VisProg (Gupta & Kembhavi, 2023), and VADAR (Marsili et al., 2025). These comparisons contrast our proposed annotation-free training framework with competing training-free program synthesis methods.
(4) Direct-answer VLMs: Proprietary and open-source VLMs directly predict answers from (image, query) pairs. This comparison contrasts VLM prediction with our reasoning and tool use paradigm.

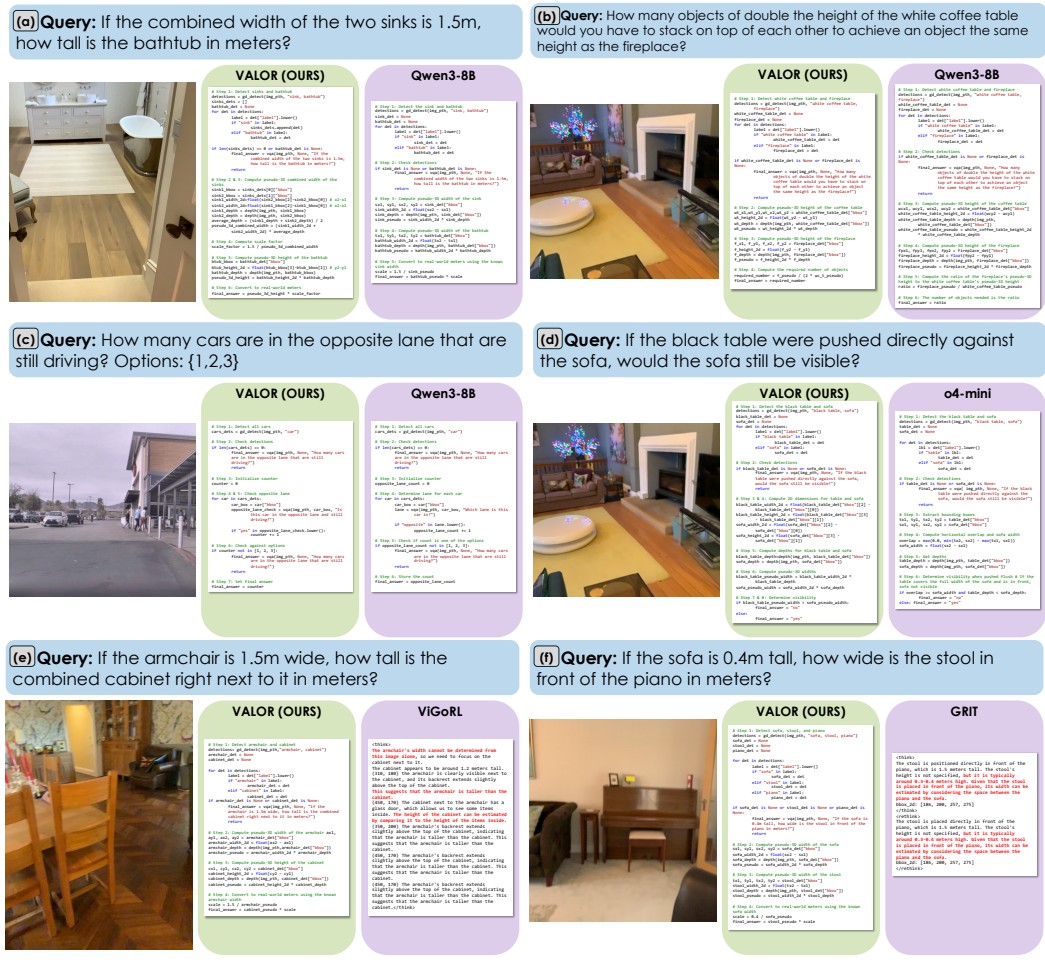

Figure 4: **Outputs for VALOR, LLMs with tools, and RL-tuned VLMs.** For each example, we show the image, query, and model output. We recommend zooming in to read the outputs.

## 4.1 VALOR VS. LLMS WITH TOOL USE

Table 1 compares VALOR against proprietary and open-source LLMs. All methods receive a query and generate plans and programs with their respective language-only LLM and access to identical vision-specialist APIs as our method. Since the baselines rely solely on language-only LLMs, there is no risk of data leakage between models and our evaluation benchmarks. All baselines, apart from the last row (VALOR), use the same visual grounding modules. We summarize our findings below:

**VALOR-RL vs open-source LLMs: The impact of verifier-improved reasoning.** Among open-source LLMs (LLama-3.2-11B, Gemma-3-12B, Qwen3-8B), Qwen3-8B performs best. In turn, VALOR-RL, our RL training approach on Qwen3-8B, shows gains over the base model: +6.4% on OMNI3D-BENCH (43.9% vs. 37.5%), +3.4% on BLINK (67.3% vs. 63.9%), +2.1% on VSR (70.3% vs. 68.2%). Both use the same specialists, isolating contributions from training with the LLM verifier. On counting tasks TALLYQA and COUNTBENCHQA (e.g., "How many people are there?") and simpler sets like GQA (e.g., "Who is wearing the dress?"), reasoning is less critical, and VALOR-RL matches Qwen3-8B. Fig. 4(a)-(c) compares programs from VALOR and Qwen3-8B. Qwen3-8B misses key details – ignoring two distinct sinks in (a), omitting the doubled height of the coffee table in (b), and overlooking the "still driving" constraint in (c) – which VALOR captures. Example (c) also shows VALOR's superior tool use, framing VQA as yes/no rather than open-ended.

**VALOR vs VALOR-RL: The impact of verifier-improved visual grounding.** Table 1 compares our full method, VALOR, that improves visual grounding via multimodal verifiers. VALOR and VALOR-RL share programs, isolating contributions from our improved visual grounding model

| | BASE MODEL | OMNI3D-BENCH | ROBOSPATIAL | BLINK | VSR | REALWORLD QA | GQA | TALLY QA | COUNTBENCH QA |
|---|---|---|---|---|---|---|---|---|---|
| *RL-Tuned VLMs* | | | | | | | | | |
| GRIT | Qwen2.5-VL-3B | 27.3 | 58.8 | **70.3** | 77.6 | 55.6 | **75.0** | 46.4 | 68.6 |
| ViGoRL | Qwen2.5-VL-7B | 13.2 | 64.9 | 68.4 | **79.1** | 52.8 | 52.5 | 49.3 | 74.5 |
| *Ours* | | | | | | | | | |
| **VALOR-RL** | Qwen3-8B | 43.9 | 61.8 | 67.3 | 70.3 | 53.5 | 57.6 | 49.5 | 67.6 |
| **VALOR** | Qwen3-8B | **44.0** | **69.5** | 69.2 | 75.6 | **57.3** | 64.4 | **51.0** | 75.9 |

Table 2: **VALOR vs RL-tuned VLMs.** GRIT (Fan et al., 2025) and ViGoRL (Sarch et al., 2025) rely on labeled data and ground in text, whereas VALOR trains without labels and leverages tools. VALOR outperforms them on reasoning-heavy spatial tasks (OMNI3D-BENCH, ROBOSPATIAL).

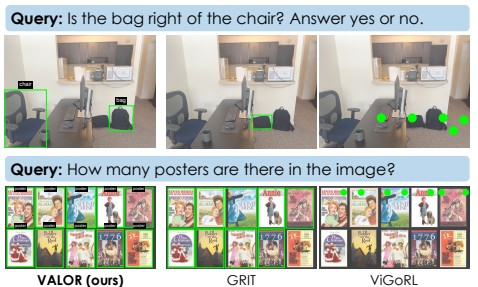

Figure 5: Visual grounding in VALOR, GRIT and ViGoRL.

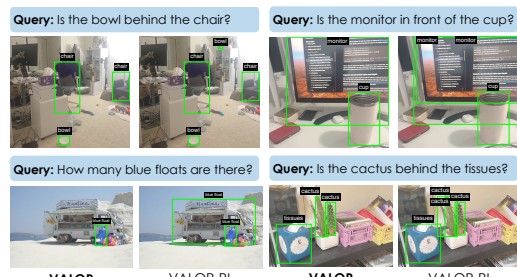

Figure 6: **VALOR vs VALOR-RL.** Multimodal verifiers improve grounding.

(§3.2). VALOR yields strong gains, particularly on grounding-focused benchmarks: +8.3% on COUNTBENCHQA, +7.7% on ROBOSPATIAL, and +5.3% on VSR. Improvements on OMNI3D-BENCH are smaller, as complex queries make reasoning the main challenge for smaller LLMs as we show in Fig. 8 in Appendix A.1.1. Fig. 6 shows our verifier-trained object detector in VALOR correcting the overpredictions made by the original GroundingDINO in VALOR-RL. Notably, improving visual grounding for spatial reasoning does not harm general object detection; our training slightly boosts performance on the COCO (Lin et al., 2014) validation set: 48.4% to 48.7% mAP. To further explore the impact of verifier-improved grounding, we include comparisons of the baseline LLMs with the improved detector in Appendix A.7. We illustrate failure cases in Appendix A.1.2.

**VALOR-RL vs proprietary LLMs.** Although not our primary focus, we compare against proprietary LLMs with access to the same specialists (top of Table 1). These LLMs are far larger or were distilled from larger models compared to open-source variants; with few details about them known. VALOR-RL, with a much smaller LLM, performs favorably, e.g., in VSR (70.3 vs 68.5% from o4-mini and Gemini-2.5-Flash) and OMNI3D-BENCH (43.9% vs 43.6% from Claude-3.5-Haiku).

Fig. 4(d) compares our method to o4-mini, the most favorable proprietary LLM, on a query requiring reasoning over size-based visibility if a table were positioned in front of the sofa. o4-mini computes the overlap between the 2D bounding boxes and separately their depth ordering, essentially assessing *current* visibility. In contrast, VALOR incorporates depth into 2D measurements to estimate 3D object widths to correctly determine visibility under the given hypothesis.

## 4.2 VALOR VS. RL-TUNED VLMS

We compare VALOR to RL-tuned VLMs GRIT (Fan et al., 2025) and ViGoRL (Sarch et al., 2025). Unlike VALOR, these models (i) reason in text without tools and (ii) require labeled data while ours is label-free. GRIT trains on TALLYQA and VSR, while ViGoRL trains on the synthetic SAT2 dataset (Ray et al., 2024); both use Qwen2.5-VL as their base VLM. Little is known about Qwen2.5-VL's training data or safeguards against overlap with multimodal spatial reasoning benchmarks, so leakage is a concern for these baselines. This is not the case for VALOR, which uses language-only LLMs trained on language-only tasks.

Table 2 shows VALOR outperforms GRIT and ViGoRL on OMNI3D-BENCH (44.0% vs 27.3% from GRIT) and ROBOSPATIAL (69.5% vs 64.9% from ViGoRL). Notably, VALOR outperforms GRIT on TALLYQA (51.0% vs 46.4%) despite GRIT training on its train set and our method being trained without labels. Fig. 4(e)-(f) shows reasining outputs from GRIT and ViGORL that reveal they ignore

grounding measurements: GRIT uses prior knowledge of stools in (e), ViGoRL estimates the cabinet height from nearby objects in (f). In contrast, VALOR correctly uses grounding information to scale object dimensions per the query.

**Tool use yields explicit grounding.** A key difference of these approaches lies in their visual grounding. GRIT predicts bounding boxes and ViGoRL points of interest through text generation which steer model predictions. In contrast, VALOR explicitly invokes specialized visual grounding models and uses the grounding output to compute object dimensions and attributes. This difference creates notable discrepancies in grounding ability, shown in Fig. 5. In the top example, GRIT omits query-relevant objects like the chair. In the bottom example, it omits the bottom-right poster. ViGoRL's point predictions are not aligned with target objects. In contrast, VALOR detects all relevant objects and grounds its reasoning accordingly.

## 4.3 VALOR VS. PROGRAM SYNTHESIS METHODS

We compare to visual program synthesis methods, Vis-Prog (Gupta & Kembhavi, 2023), ViperGPT (Surís et al., 2023), and VADAR (Marsili et al., 2025), which produce executable programs for spatial reasoning. These methods use proprietary LLMs, such as GPT-4o, with sophisticated prompting to generate programs. VALOR tunes a much smaller Qwen3-8B base model. All methods are annotation-free, but VALOR trains for reasoning and visual grounding with the help of verifiers.

| | BASE MODEL | OMNI3D-BENCH | GQA |
|---|---|---|---|
| VisProg | GPT-4o | 17.8 | 46.9 |
| ViperGPT | GPT-3.5 | 25.5 | 42.0 |
| VADAR | GPT-4o | 38.9 | 46.1 |
| **VALOR-RL** | Qwen3-8B | 43.9 | 55.0 |
| **VALOR** | Qwen3-8B | **44.0** | **63.0** |

Table 3: VALOR vs Program Synthesis.

Table 3 shows results on OMNI3D-BENCH and GQA. We evaluate on OMNI3D-BENCH's 100 held out queries and the GQA subset from VADAR (Marsili et al., 2025). VALOR outperforms all methods on OMNI3D-BENCH (44.0% vs 38.9% from VADAR) and GQA (63.0% vs 46.9% from VisProg), despite using a smaller open-source LLM than GPT-based baselines.

## 4.4 VALOR VS. VLMS

We compare VALOR to proprietary and open-source VLMs that predict answers directly from (image, query) pairs. As these are trained on proprietary datasets, data leakage is a concern; thus, we evaluate on benchmarks released after their release. Table 4 reports results on OMNI3D-BENCH, ROBOSPATIAL, and COUNTBENCHQA. Although the benchmarks predate the model release, we include comparisons to GPT-5-mini in Appendix A.5.

| | OMNI3D-BENCH | ROBOSPATIAL | COUNTBENCH QA |
|---|---|---|---|
| *Proprietary Models* | | | |
| GPT-4o | 35.0 | 57.9 | 84.7 |
| Gemini-2.0-Flash | 34.6 | 60.9 | **88.6** |
| Claude-3.5-Haiku | 32.5 | 59.6 | 77.6 |
| *Open-Source Models* | | | |
| Llama3.2-11B | 22.7 | 39.9 | 71.5 |
| Qwen2.5-VL-7B | 20.7 | 60.1 | 82.5 |
| **VALOR-RL** | 43.9 | 61.8 | 67.6 |
| **VALOR** | **44.0** | **69.5** | 75.9 |

Table 4: VALOR vs Direct Prediction VLMs.

We observe that VALOR outperforms larger proprietary models on complex reasoning benchmarks OMNI3D-BENCH (44.0% vs 35.0% from GPT-4o) and ROBOSPATIAL (69.5% vs 60.9% from Gemini-2.0-Flash), demonstrating the effectiveness of our approach. These improvements are larger when comparing to open-source models: VALOR achieves +21.3% from Llama3.2 on OMNI3D-BENCH (44.0% vs 22.7%). On COUNTBENCHQA, VALOR trails. This suggests that VLMs may be stronger bases for visual grounding than current state-of-the-art models like GroundingDINO.

## 4.5 IMPACT OF TRAINING DATA SIZE

Our proposed annotation-free training framework scales inherently. We analyze how the number of training samples impacts both reasoning and visual grounding capabilities in VALOR.

**Impact of training set size on reasoning.** Table 5 shows VALOR-RL performance on OMNI3D-BENCH, our most challenging spatial reasoning benchmark, as training set size scales.

| | Training Samples | OMNI3D-BENCH |
|---|---|---|
| Qwen3-8B | | 37.5 |
| VALOR-RL | 40 | 40.0 |
| VALOR-RL | 160 | 39.2 |
| VALOR-RL | 400 | 40.8 |
| VALOR-RL | 800 | 43.9 |

Table 5: Effect of training set size for reasoning.

We tune Qwen3-8B with $N = \{40, 160, 400, 800\}$ queries (half from OMNI3D-BENCH, half generated), and also report results for the base model. The VALOR-RL variant reported in experiments trains on 800 samples. Specialist models are fixed across all variants. We find VALOR surpasses the base Qwen3-8B model with as few as 40 samples (40.0% vs 37.5%), and continues improving with more data (43.9% with 800 samples vs 40.8% with 400 samples). This demonstrates strong data scalability.

**Impact of training set size on visual grounding.** Fig. 7 shows the impact of training data on verifier-trained visual grounding. We tune GroundingDINO with $M = \{5.6k, 8.1k, 17.4k, 30.8k\}$ pseudo-annotations and plot accuracy on RO-BOSPATIAL, VSR, and COUNTBENCHQA – benchmarks which heavily rely on visual grounding. All variants execute identical programs, so performance differences arise solely from grounding. The upward trend as training data increases demonstrates the scalability of our verifier-guided training.

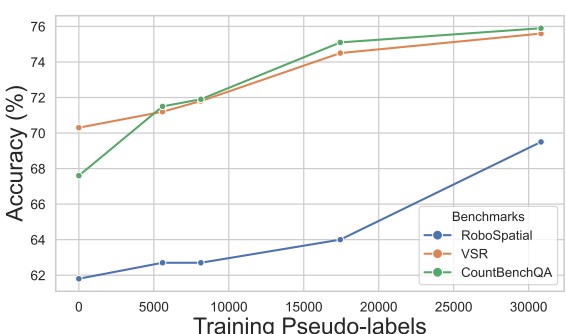

Figure 7: Effect of training data for visual grounding.

### 4.6 SUPERVISED FINE-TUNING VS. REINFORCEMENT LEARNING

Although not the primary focus of our work, we ablate the post-training algorithm by comparing our verifier-based RL training (§ 3.1) with supervised fine-tuning (SFT) in Table 6. Our method optimizes a sparse verifier reward via reinforcement, whereas SFT provides dense supervision over full model outputs. To keep provenance of outputs and verifier signal consistent, we construct an SFT dataset by sampling VALOR training outputs with verifier reward $R \geq 0.7$, ensuring both methods learn from the same verifier-filtered program traces.

|  | OMNI3D-BENCH | ROBOSPATIAL | COUNTBENCH QA |
|---|---|---|---|
| SFT | 38.3 | 64.5 | 74.5 |
| VALOR | 44.0 | 69.5 | 75.9 |

Table 6: **SFT vs RL.** We tune Qwen3-8B on high-reward samples with SFT and compare to our RL-based VALOR.

We fine-tune Qwen3-8B on these samples using SFT and evaluate the resulting model with the improved visual grounding module for a direct comparison to VALOR. SFT consistently underperforms RL on reasoning-heavy benchmarks such as OMNI3D-BENCH (38.3% vs 44.0%) and ROBOSPATIAL (64.5% vs 69.5%), indicating that GRPO more effectively improves model reasoning. On the counting benchmark COUNTBENCHQA, both approaches perform similarly. We include all evaluation results in Table 9 in Appendix A.8.

## 5 CONCLUSION & FUTURE WORK

We introduce VALOR, an annotation-free training paradigm for visual reasoning that leverages multimodal verifiers to improve LLM reasoning and visual grounding. A key insight driving our approach is that stronger VLMs/LLMs are often more reliable as verifiers than generators, motivating a verifier-based strategy for improving visual reasoning. We evaluate VALOR on visual reasoning tasks spanning 2D and 3D image understanding to object counting, highlighting success and failures through an extensive quantitative analysis. Further failure analysis is in Appendix A.1, Fig. 8 and 9. We release code and models to facilitate future work. Some promising directions are below:

- We show how multimodal verifiers can pseudo-annotate images and improve grounding. Integrating them into RL training is an extension of this paradigm.
- Harvesting hard negatives with VLM verifiers improved visual grounding models. Following similar principles for reasoning through guided query generation is a promising direction.
- VLMs are strong grounders, with Gemini-2.0-Flash achieving 88.6% on COUNTBENCHQA via direct prediction. Adapting them as grounding tools could further benefit methods like VALOR.

ACKNOWLEDGMENTS

We thank Aadarsh Sahoo, Ilona Demler, and Ziqi Ma for their feedback on the project. The project is funded by Meta through the LLM evaluation research grant and partly through Caltech's CAST program. We also thank Google's Gemma Academic program for granting us API credits for their LLMs.

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

# A APPENDIX

## CONTENTS

## A.1 FAILURE CASES

In this section, we show error cases of VALOR. We include errors in reasoning in §A.1.1, along with a brief explanation on why the program is incorrect. In §A.1.2 we highlight errors in grounding.

## A.1.1 REASONING FAILURE CASES

We show reasoning failure cases in Fig. 8. In the query on the left, we find VALOR oversimplifies the challenging hypothetical posed in the query by ignoring the direction of travel specified for the chalice, and performing a comparison of depth over the wrong objects. In the second query on the right, VALOR incorrectly conflates depth with distance, while this query requires considering distance in 3D (the utensil holder is directly below the drawer and directly to the right of the edge of the counter). Additionally, VALOR omits comparisons with the drawers. Despite improved performance, the challenging hypotheticals in OMNI3D-BENCH remain difficult for VALOR.

## A.1.2 GROUNDING FAILURE CASES

We show grounding failure cases of VALOR in Fig. 9. We find that while VALOR improves grounding, under predictions remain: in the first example VALOR finds 5 toilets as opposed to 6 and only 2 of the 5 basins. In the second example, VALOR does not successfully ground all of the windows of the bus. Lastly, some duplicate predictions remain, as shown in the duplicate detection of the glass.

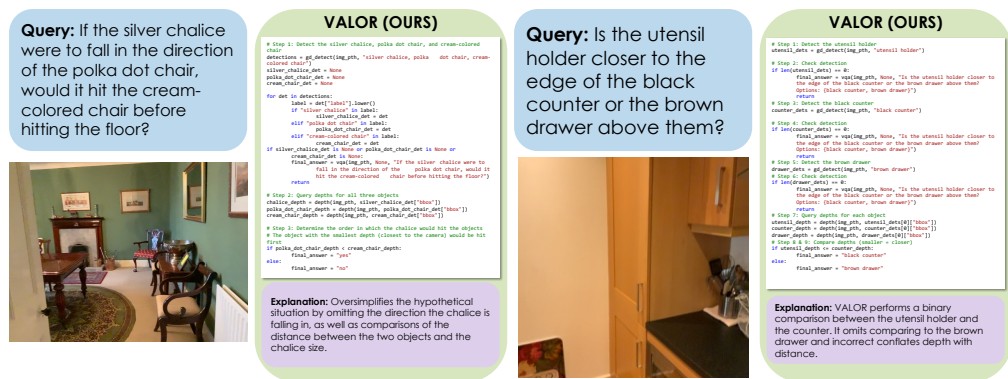

Figure 8: Reasoning failure cases of VALOR on challenging hypothetical examples from OMNI3D-BENCH. For each query, we show the image and generated program, as well as an explanation of incorrectness.

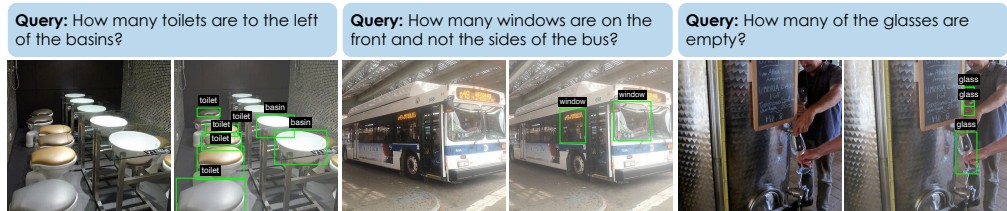

Figure 9: Grounding failure cases of VALOR. For each query we show the original image and the outputs of the verifier-tuned grounding model.

## A.2  ADDITIONAL VISION SPECIALIST DETAILS

Our vision specialist API consists of an object detector, a segmentation model, a depth estimation model, and a VQA model. Fig 16 shows the full API docstring. We provide additional details of each specialist model below.

**Object Detection.** We take a GroundingDINO-T (Liu et al., 2023b) model pretrained on Objects365 (Shao et al., 2019), GoldG (Kamath et al., 2021), and Cap4M (Li et al., 2022) as our object detector. We specify in our API that the prompt to the object detector can either be comma-separated nouns, e.g., "fireplace, coffee table, sofa", or nouns accompanied by an appearance attribute, e.g., "black coffee table". We additionally specify that spatial relationship attributes should not be included in the prompt, as we found the object detector unable to consider them appropriately.

**Depth Estimation.** For depth estimation we use the `vit-l-normal` checkpoint from MoGe2 (Wang et al., 2025). Our depth module takes as input an object bounding box and returns the estimated depth at the center of the box.

**VQA.** The primary purpose of the VQA module is to answer queries about the appearance of an object specified by an input bounding box (e.g. "What color is this object?"). We take a crop of the image around the input bounding box and pass the cropped image to GPT-5-mini (OpenAI, 2025a) alongside the query. We also allow broader questions in the event of an early exit due to failed detections, specified by a bounding box of `None` (see Fig. 17). In this case, the entire image is passed to GPT-5-mini.

## A.3  ADDITIONAL EVALUATION DATA DETAILS

We provide additional details on the datasets used for evaluation below. We report exact-match accuracy for all yes/no, multiple choice, and integer response queries, and MRA for floating point response queries following Marsili et al. (2025). Additional details on the evaluation datasets can be found in Appendix A.3. To account for synonyms in the open-ended answers of GQA, we use GPT-5-mini (OpenAI, 2025a) as a judge for that benchmark. The prompt to GPT-5-mini is in Figure 20.

**Omni3D-Bench** is a VQA dataset focused on 3D grounding and reasoning (Marsili et al., 2025). The dataset includes 500 challenging (image, question, answer) tuples of diverse real-world scenes sourced from Omni3D (Brazil et al., 2023). Examples include *"If the 3D height of the two-seat sofa is 0.50 meters, what is the 3D height of the dining table in meters?"* and *"Which object is closer to the fireplace: the sofa or the white coffee table?"*. We keep the first 100 samples in OMNI3D-BENCH for evaluation and use the remaining 400 samples for training.

**TallyQA** is a VQA dataset that exclusively tests counting (Acharya et al., 2019). Examples include *"How many people are in this picture?"* and *"How many sheep do you see?"*. We evaluate on a subset of 491 questions provided in GRIT (Fan et al., 2025) which are uniformly sampled to ensure an equal distribution of ground truth object counts from 0 to 9.

**GQA** is a popular visual reasoning dataset where queries are generated using scene graphs (Hudson & Manning, 2019). Queries in GQA primarily pertain to 2D spatial relationships and the visual appearance of objects. Examples include *"Who is dressed in yellow?"* and *"What is the name of the animal to the left of the bookcase?"*. We evaluate on a subset of 509 questions proposed in GRIT (Fan et al., 2025), which has been manually filtered to remove ambiguous answers.

**RoboSpatial** is a large-scale VQA dataset of indoor and tabletop environments for spatial reasoning in robotics (Song et al., 2025). Queries in ROBOSPATIAL are procedurally generated from 3D scans. Examples include *"Can the chair be placed in front of the table?"* and *"Is the mug to the left of the laptop?"*. We evaluate on the relevant `compatibility` and `configuration` tasks.

**VSR** is a spatial reasoning dataset composed of images from COCO (Lin et al., 2014) annotated with spatial relationships between two objects in the image (Liu et al., 2023a). Queries are formatted as statements which a model must evaluate as `True` or `False`. Examples include *"The potted plant is at the right side of the bench"* and *"The cow is ahead of the person"*.

**BLINK** is a benchmark of 14 visual reasoning challenges that can are trivial for humans but challenging for VLMs (Fu et al., 2024). The dataset is composed of single-image queries and multi-image queries. We restrict our evaluation to single-image queries and evaluate on the `val` split of the relevant `counting` and `spatial relationship` tasks.

**CountBenchQA** is a counting benchmark released with the PaliGemma model (Beyer et al., 2024). The dataset consists of manually annotated counting queries with images sourced from the LAION-400M dataset (Schuhmann et al., 2021). Examples include *"How many light bulbs are there in the image?* and *"How many food containers are there in the image?"*. We evaluate on the publicly available 491 image split.

**RealWorldQA** is a dataset of 765 VQA queries sourced from diverse real-world scenes (xAI, 2024). The benchmark tests general spatial understanding with an emphasis on real-world outdoor scenes. Examples include *"Is there a bike lane to my right? Please answer directly with a single word or number."* and *"How many pedestrians are there within 50 meters from us? A. 3 B. 5 C. 7 or more"*. We evaluate on the full `test` split.

## A.4 GENERATED TRAINING DATA

Our approach involves training both a base LLM for logic and program decomposition as well as a visual grounding model. In both cases, we leverage our method's lack of reliance on ground truth labels to generate supplementary synthetic data used for training. In this section we provide additional details on this process.

### A.4.1 GENERATING SPATIAL REASONING QUERIES FOR IMPROVED REASONING.

Since our method requires only spatial reasoning queries without labels, we can scale training beyond small labeled datasets by sampling images from SA-1B (Kirillov et al., 2023) and prompting Gemini-2.5-Flash to inspect each image for suitability (e.g., multiple objects for grounding). For suitable images, we prompt it to generate five queries per image and uniformly select one. While we don't use these images to train our LLM, seeding query generation with reference images produces more diverse spatial reasoning queries and ensures referenced objects are in-distribution. Similarly,

sampling from five predictions per image yields greater diversity than generating single queries directly. We show example queries with their reference images in Figure 10 and include the generation prompt in Figure 19. We use default hyperparameters for Gemini.

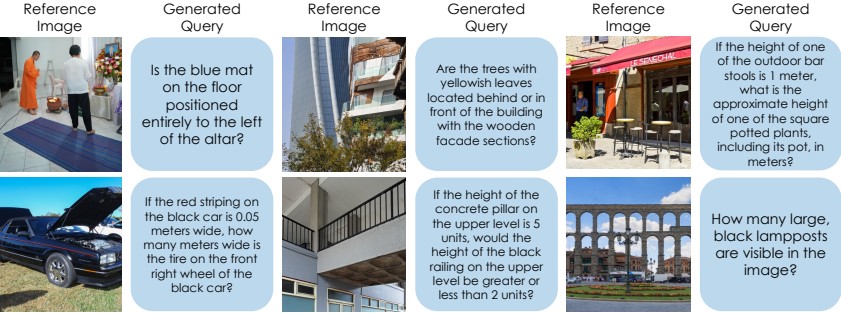

Figure 10: Example spatial reasoning queries generated from reference images.

### A.4.2 GENERATING DATA FOR IMPROVED VISUAL GROUNDING

We use spatial reasoning queries to generate detection signals through VLM verifiers to capture the distribution of object types and target domains pertinent to spatial reasoning. We source (image,query) pairs from spatial reasoning benchmarks TALLYQA, GQA, VSR and OMNI3D-BENCH, and supplement these with queries generated as described by §A.4.1. W

From these (image,query) pairs, we use our language-based reasoning model to produce a plan and code. We subsequently extract all visual grounding queries in the program (GD_DETECT) and execute them using a GroundingDINO model with both BOX_THRESHOLD and TEXT_THRESHOLD lowered to 0.2 to ensure higher recall. Lowering the thresholds leads to consistent overprediction of bounding boxes, which we then filter using a three step process using GPT-5-mini (OpenAI, 2025a) as our VLM verifier. We use default generation hyperparameters for GPT-5-mini. All prompts used for verification are in Fig. 21. We detail verification step below:

1. **Coarse filtering:** We overlay all predicted boxes on the original image and pass both the annotated and original images to GPT-5-mini. Each box is numbered and labeled, with a JSON of predictions provided to the model. GPT-5-mini is prompted to remove boxes whose content doesn't match the predicted label or that are poorly fitted. This coarse filtering removes a significant portion of incorrect boxes.

2. **Per-crop object check:** For each remaining box, we crop the image around the box and upscale it so the longest side is 640px while preserving aspect ratio. We pass each crop with its predicted label to GPT-5-mini and ask whether the primary object matches the label. We discard boxes that GPT-5-mini flags as incorrect.

3. **Deduplication:** As a final step, we repeat the process from Step 1 with all remaining boxes. GPT-5-mini is prompted to focus on duplicates, as these likely weren't filtered in the second stage due to being semantically correct. We remove boxes flagged as duplicates and keep the remaining boxes as our final pseudo-labels.

We show additional samples from our VLM-based verifier in Figure 11. For each sample, we show the original image, the bounding boxes predicted by the GroundingDINO model with reduced confidence threshold and the final verifier outputs. We overlay all bounding box predictions in red along with the predicted label. The low-threshold predictions are the input to our verification pipeline (step 1 above). We observe that despite a highly cluttered input image, our three-stage verification pipeline is effective at removing semantically incorrect and duplicate boxes.

**Verifier Failures.** In Figure 12, we visualize failure cases for our VLM verifier. For each sample, we show the original image, the boxes predicted by the GroundingDINO model with reduced confidence threshold and the final verifier outputs. From Figure 12, we observe that our verifier occasionally fails when two semantically identical objects overlap significantly. This is highlighted in the discarded boxes on the right side of the truck in the top-left example, the crowd of people in the bottom left example, and the two men on the motorcycle in the bottom right. On the top left, the

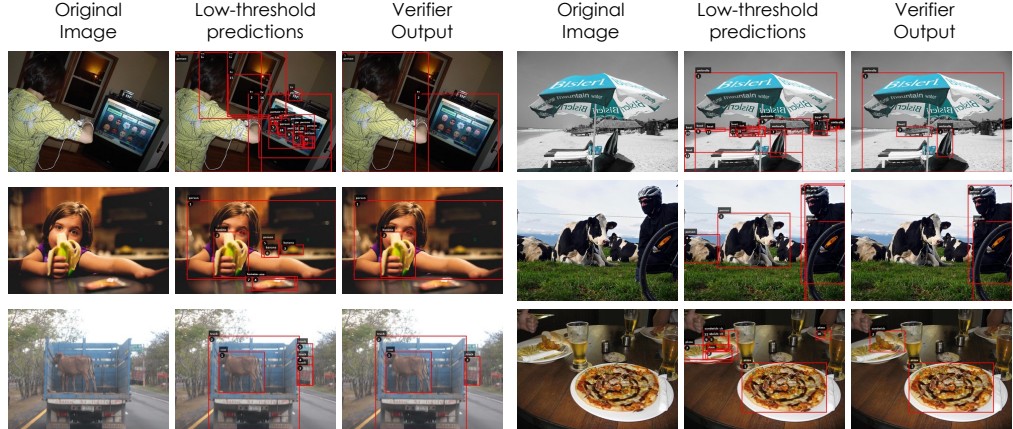

Figure 11: Additional examples of VLM-based verification of visual grounding model outputs. For each sample we show the original image, the predicted bounding boxes by the low-threshold model, and the final pseudo-annotations produced by the VLM verifier.

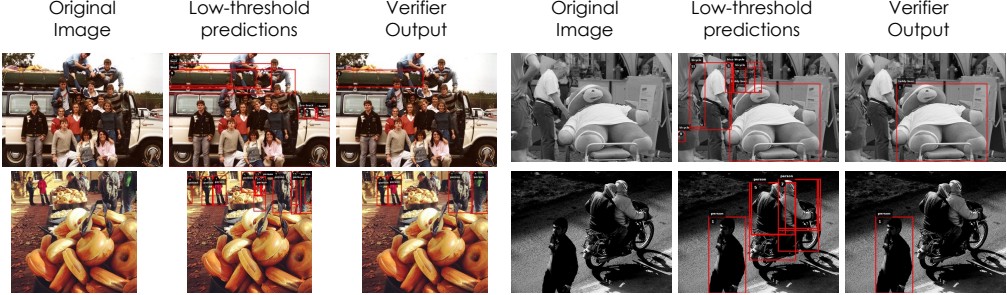

Figure 12: Failure cases of our VLM-based verifier. For each sample we show the original image, the predicted bounding boxes by the low-threshold model, and the final pseudo-annotations produced by the VLM verifier. Our verifier occasionally fails when two similar objects overlap significantly.

small bicycle bounding boxes are discarded by the per-crop semantic check, likely due to being too small to accurately resolve. Verification in these settings remains a challenge.

## A.5 VALOR VS GPT-5-MINI

We report direct prediction accuracy of GPT-5-mini in Table 7. We note that GPT-5-mini was released after all of the benchmarks, so we cannot rule out data leakage, as it is possible these benchmarks were included in its training corpus. Additionally, to explore how object grounding affects performance for GPT-5-mini, we report an additional variant GPT-5-mini (grounded) where the model is prompted to explicitly predict bounding boxes for the relevant objects in the query prior to a final response.

Despite its scale and capabilities, VALOR outperforms GPT-5-mini on OMNI3D-BENCH, our most reasoning-intensive benchmark. Notably, VALOR – built on a much smaller 8B language-only model – achieves strong results across all benchmarks, which underscores the effectiveness of our verifier-guided training framework. The grounded variant of GPT-5-mini performs similarly to the direct-prediction version on most tasks but degrades on several reasoning-intensive benchmarks, improving only on the counting datasets (TALLYQA, COUNTBENCHQA). This suggests that while explicit grounding can help VLMs in object-centric counting scenarios, visually grounded multi-step reasoning without tool-use remains challenging even for large proprietary VLMs.

Figure 13 additionally compares the visual grounding produced by VALOR with that of GPT-5-mini, the VLM we use as a verifier. Although GPT-5-mini is highly effective at evaluating object detections, we observe that it often struggles when generating bounding boxes for spatial queries.

| | OMNI3D-BENCH | ROBOSPATIAL | BLINK | VSR | REALWORLD QA | GQA | TALLY QA | COUNTBENCH QA |
|---|---|---|---|---|---|---|---|---|
| GPT-5-mini | 40.9 | 76.3 | 81.0 | 84.4 | 62.2 | 81.9 | 55.2 | 84.3 |
| GPT-5-mini (grounded) | 35.5 | 76.2 | 76.0 | 82.4 | 57.4 | 80.4 | 55.4 | 84.7 |
| **VALOR** | **44.0** | **69.5** | 69.2 | 75.6 | **57.3** | 64.4 | 51.0 | **75.9** |

Table 7: VALOR vs GPT-5-mini direct prediction. For the (grounded) variant, GPT-5-mini is prompted to predict bounding boxes for all relevant objects prior to responding.

As shown in the figure, GPT-5-mini frequently outputs misaligned or overly large boxes, failing to localize key objects reliably. In contrast, VALOR produces accurate and well-localized detections. These examples underscore an important distinction between verification and generation abilities: a VLM can provide reliable binary judgments about correctness even when its own direct grounding predictions are imperfect.

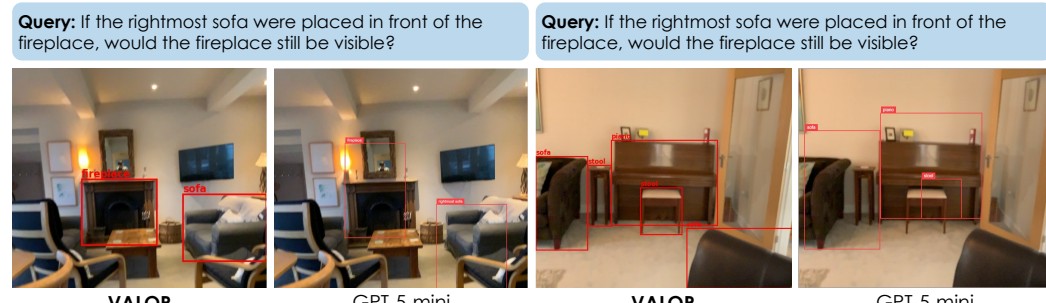

Figure 13: Visual grounding comparison of VALOR vs GPT-5-mini. Despite serving as a powerful VLM verifier, GPT-5-mini struggles to accurately ground important objects, often predicting poorly placed and overly large boxes.

### A.6 ADDITIONAL TRAINING DETAILS

In this section we provide additional details on VALOR's training. Specifically, §A.6.2 includes further detail on our reward models for LLM reasoning, §A.6.3 provides details on RL for LLM reasoning training and §A.6.4 includes details on verifier-based visual grounding model training.

### A.6.1 GRPO.

We provide an overview of our optimization algorithm, Group Relative Policy Optimization (GRPO; Guo et al. (2025)) below.

**Setup.** Given a query $q$, our base LLM $\pi_\theta$, parametrized by $\theta$, generates a natural language plan $p$ and corresponding Python code $c$ with access to predefined tools via their API $\mathcal{A}$: $(p, c) = \pi_\theta(q; \mathcal{A})$. We aim to find a model, $\pi_{\theta^*}$, that optimizes program correctness, API usage and coordination.

**Optimization.** We optimize our LLM $\pi_\theta$ using GRPO; Guo et al. (2025). For each query $q$, we sample $G$ responses $\mathcal{O} = \{(q, p^{(i)}, c^{(i)})\}_{i=1}^{G}$ and compute centralized advantages:

$$A^{(i)} = R(q, p^{(i)}, c^{(i)}) - \frac{1}{G}\sum_{j=1}^{G} R(q, p^{(j)}, c^{(j)}) \quad (2)$$

The GRPO objective maximizes expected advantages while maintaining policy stability:

$$\mathcal{L}_{\text{GRPO}}(\theta) = \frac{1}{G}\sum_{i=1}^{G}\left( \min\left( s^{(i)}A^{(i)}, \text{clip}\left( s^{(i)}, 1-\epsilon, 1+\epsilon \right)A^{(i)} \right) - \beta\text{KL}[\pi_\theta || \pi_{\text{ref}}] \right) \quad (3)$$

where $s^{(i)} = \frac{\pi_\theta(p^{(i)}, c^{(i)}|q)}{\pi_{\text{old}}(p^{(i)}, c^{(i)}|q)}$ is the importance ratio, $\epsilon = 0.2$ is the clipping parameter and $\beta = 0.01$ is the KL penalty coefficient. This formulation encourages improved program synthesis quality while preventing drift from the pre-trained base policy $\pi_{\text{ref}}$.

### A.6.2 REWARD DESIGN.

We design a reward model powered by LLMs which verify if predicted programs are correct. Given a query $q$, our policy LLM is tasked with predicting a natural language plan $p$ and Python program $c$. Our final reward function for these outputs is given as:

$$R(q, p, c) = r_{\mathrm{fmt}}(p, c) \cdot \left[ \lambda_{\mathrm{sn}} \, r_{\mathrm{sn}}(c) + \lambda_{\mathrm{log}} \, r_{\mathrm{log}}(q, p) + \lambda_{\mathrm{att}} \, r_{\mathrm{att}}(q, p) + \lambda_{\mathrm{sp}} \, r_{\mathrm{sp}}(q, p) + \lambda_{\mathrm{ad}} \, r_{\mathrm{ad}}(p, c) \right] \tag{4}$$

We describe each reward function in further detail below. For all LLM-based rewards (logic, attribute, spatial, and adherence), we use a frozen Gemini-2.5-Flash LLM as our verifier. All prompts to LLM verifiers can be found in Figure 18.

**Format Reward ($r_{\mathrm{fmt}}$):** We employ a format reward $r_{\mathrm{fmt}}$ to ensure the LLM predicts parsable answers in the expected `<plan></plan><answer></answer>`. Given an LLM output, if either of the tags are missing, or the output inside them is empty, the reward returns 0. Otherwise, the reward returns 1. This reward acts as a hard constraint and ensures the model is only rewarded for properly formatted responses.

**Syntax Reward ($r_{\mathrm{sn}}$):** Our syntax reward $r_{\mathrm{sn}}(c)$ checks whether the predicted program $c$ executes without Python errors. As we are evaluating syntax and not execution accuracy, we replace all calls to vision specialist models with dummy functions to lower runtime. The reward is 1 for all programs that execute without error and 0 otherwise. Since we found pre-trained LLMs to be proficient at writing code, we lower the weight of this reward to $\lambda_{\mathrm{sn}} = 0.1$.

**Logic Reward ($r_{\mathrm{log}}$):** We use an LLM verifier to act as a critic over the logical correctness of the predicted plan. Importantly, the critic LLM is only given the query and the plan, not the code, as we found it more adept at judging the logical correctness of a plan in natural language as opposed to a program. The critic LLM is prompted to return a binary decision of 1 if the plan is reasonable and sequentially coherent for the query or 0 otherwise. We found logical correctness to be the largest source of error in pre-trained LLMs, leading us to raise the weight of this reward to $\lambda_{\mathrm{log}} = 0.3$.

**Attribute Reward ($r_{\mathrm{att}}$):** We task an LLM verifier with ensuring all object properties (height, color, etc.) mentioned in the query are correctly identified in the plan and the plan details how to compute them correctly. We found this reward particularly important on OMNI3D-BENCH to ensure object dimensions are computed in a 3D-aware manner. We weigh this reward $\lambda_{\mathrm{att}} = 0.2$.

**Spatial Reward ($r_{\mathrm{sp}}$):** We found the omission of spatial relationships highlighted in the query to be a significant source of error among pre-trained LLMs (e.g. treating "leftmost sofa" as "sofa'). Thus, we invoke an LLM verifier to ensure all spatial relationships detailed in the query are explicitly and correctly addressed in the plan. We set $\lambda_{\mathrm{sp}} = 0.2$.

**Adherence Reward ($r_{\mathrm{ad}}$):** As the other semantic rewards (logic, attribute, and spatial) encourage detailed and correct plans, we employ an adherence reward $r_{\mathrm{ad}}$ to ensure the predicted code implements the plan faithfully. The reward is computed by an LLM verifier that is given the plan and predicted code and prompted to return 1 if the code matches the plan exactly and 0 otherwise. We set $\lambda_{\mathrm{ad}} = 0.2$.

**Reward Model Analysis.** Each of our reward models targets a distinct aspect of spatial reasoning, with its own evaluation criteria. Figure 14 illustrates the role of each reward head evaluated by an LLM verifier – spatial, attribute, logic, and code – by showing representative training outputs that failed to achieve a full reward. By focusing on examples that were (correctly) penalized by a particular head, we highlight the types of errors that would otherwise go unaddressed in the absence of that head.

In the Spatial Reward example, the model fails to verify which potted plant is closest, an omission which the reward model detects that would otherwise lead to an incorrect answer when multiple plants are present. The Attribute Reward example demonstrates that the model does not check whether the candles are on, which is correctly detected by the reward model. In the Logic Reward panel, the reward model detects a discrepancy between the VQA prompt invoked in the plan and the original query. Finally, the Code Reward identifies that the generated program does not compare the largest table, even though the plan explicitly says to.

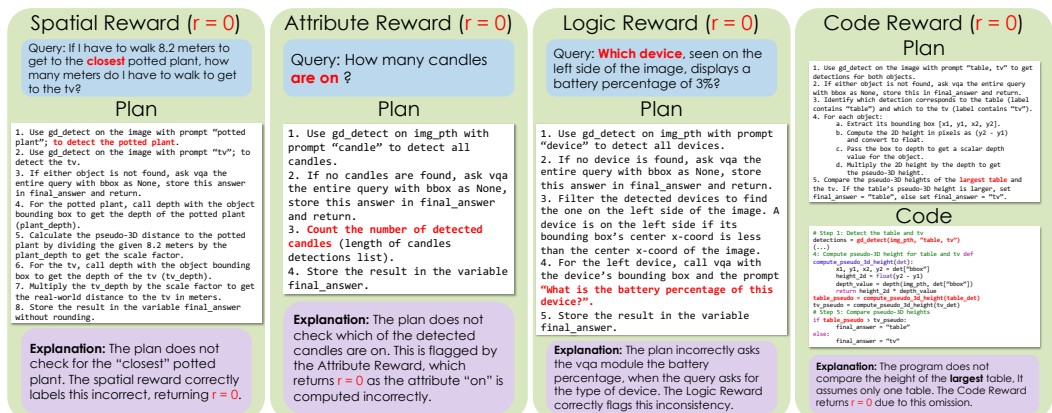

Figure 14: **Analysis of reward models.** For each reward type evaluated by an LLM verifier –spatial, attribute, logic, and code – we show a representative model output from training that failed to receive the full reward. Each panel presents the input provided to the reward model, along with a description of the specific error the reward model correctly identified. We highlight sources of error in red. We recommend zooming to read model outputs.

### A.6.3 LLM REASONING TRAINING

**Training Data.** We train our base LLM with 800 spatial reasoning queries, consisting of 400 generated queries (§A.4.1) and the last 400 queries from OMNI3D-BENCH. We reserve the first 100 queries from OMNI3D-BENCH for evaluation.

**Implementation.** We optimize a Qwen3-8B model using GRPO (Guo et al., 2025), building on the verl repository (Sheng et al., 2025). We use the system prompt shown in Fig. 15 and the reward described in §A.6.2. We set `enable_thinking` to `False` in our tokenizer as we found it equivalent to predicting the plan in `<plan></plan>` tags while reducing output length. We train on 8 A100 GPUs for 4 epochs with an batch size of 64, a micro batch size per-gpu of 1, and a group size of 5. We detail all hyperparameters in Table 10.

### A.6.4 GROUNDINGDINO FINE-TUNING

**Training Data.** For training we use data generated by our VLM-verification pipeline described in §A.4.2. We source images from the training sets of spatial reasoning benchmarks TALLYQA, GQA, VSR and OMNI3D-BENCH, as well as queries generated as described in §A.4.1. Our training dataset consists of 30,826 pseudo-annotations sourced from 7,373 images. Of these images: 124 are from OMNI3D-BENCH, 2,312 are from GQA, 1,655 are from TALLYQA, 1718 are from VSR, and 1,564 are generated from SA-1B (Kirillov et al., 2023). In our experiments, we evaluate how our verifier-based training affects GroundingDINO's performance on the validation set of COCO (Lin et al., 2014). Despite training without labels, to ensure this is a fair evaluation we discard images from TALLYQA and VSR that are sourced from the validation set of COCO.

**Implementation.** We fine-tune GroundingDINO using the Open-GroundingDINO repository (Zuwei Long, 2023). We use the `GroundingDINO-T` variant with a Swin Transformer (Liu et al., 2021) vision backbone and BERT (Devlin et al., 2019) language encoder. We keep the vision and language encoder frozen and train the rest of the model. We train on 4 A100 GPUs for 7 epochs with a batch size of 16 and learning rate of $10^{-6}$. As the number of images from our source datasets are not uniformly distributed, we implement a balanced batch sampler that samples from each source dataset with equal probability. We use a step learning rate scheduler that decreases the learning rate to $10^{-7}$ at the start of epoch 4. We detail all hyperparameters in Table 11.

| | Omni3D-Bench | RoboSpatial | BLINK | VSR | RealWorld QA | GQA | Tally QA | CountBench QA |
|---|---|---|---|---|---|---|---|---|
| *Direct Comparisons* | | | | | | | | |
| Gemma-3-12B (base GD) | 24.4 | 54.0 | **57.4** | 57.9 | **47.3** | 46.0 | 48.9 | 67.8 |
| Gemma-3-12B (tuned GD) | **25.2** | **57.0** | 55.9 | **60.9** | 44.7 | **52.1** | **49.5** | **72.7** |
| Qwen3-8B (base GD) | 37.5 | 60.5 | 63.9 | 68.2 | 55.3 | 57.4 | 50.1 | 68.6 |
| Qwen3-8B (tuned GD) | **37.9** | **66.7** | **68.8** | **72.4** | **56.2** | **63.9** | 50.1 | **74.1** |
| **VALOR-RL** (base GD) | 43.9 | 61.8 | 67.3 | 70.3 | 53.5 | 57.6 | 49.5 | 67.6 |
| **VALOR** (tuned GD) | **44.0** | **69.5** | **69.2** | **75.6** | **57.3** | **64.4** | **51.0** | **75.9** |

Table 8: **Additional evaluations of verifier-improved visual grounding.** We report the performance of baseline LLMs with the base detector (base GD) and the fine-tuned detector (tuned GD).

| | Omni3D-Bench | RoboSpatial | BLINK | VSR | RealWorld QA | GQA | Tally QA | CountBench QA |
|---|---|---|---|---|---|---|---|---|
| SFT | 38.3 | 64.5 | 67.2 | 68.2 | 56.3 | 60.5 | 49.5 | 74.5 |
| VALOR | **44.0** | **69.5** | **69.2** | **75.6** | **57.3** | **64.4** | **51.0** | **75.9** |

Table 9: **SFT vs RL.** We tune Qwen3-8B on high-reward samples with SFT and compare to our RL-based VALOR. Both models use the verifier-improved grounding module.

## A.7 ADDITIONAL EVALUATIONS OF VERIFIER-IMPROVED VISUAL GROUNDING

To additionally illustrate the impact of the verifier-improved GroundingDINO (GD; Liu et al. (2023b)) module on other models, we report the performance of Gemma-3-12B and Qwen3-8B equipped with the fine-tuned detector (tuned GD) in Table 8. These results are directly comparable to the variants reported in Table 1 – the only difference is the grounding module. We include the baselines with the pre-trained detector (base GD), VALOR-RL (that also uses the base GD) and the full VALOR (that uses the tuned GD) for completeness.

We observe that fine-tuning the visual grounding module improves performance across datasets for both open-source base LLMs (base vs tuned GD). At the same time, comparing VALOR against Qwen3 with the tuned detector shows that – particularly on benchmarks where reasoning is the core challenge (e.g., Omni3D-Bench) – improving visual grounding alone is insufficient; the verifier-guided reasoning improvements introduced by our method in Section 3.1 are important.

## A.8 SUPERVISED FINE-TUNING VS REINFORCEMENT LEARNING

We include all evaluation results for the supervised fine-tuning vs reinforcement learning ablation (§ 4.6) in Table 9. We find that VALOR outperforms SFT on reasoning-intensive benchmarks (OMNI3D-BENCH, ROBOSPATIAL), but peforms similarly on counting tasks (TALLYQA and COUNTBENCHQA).

## A.9 HYPERPARAMETERS

We include all hyperparameters used for both LLM reasoning training and GroundingDINO fine-tuning in the tables below. For inference, we use the recommended sampling parameters for `no-think` prompting of Qwen3-8B (Yang et al., 2025).

| Hyperparameter | Value |
|---|---|
| Batch size | 64 |
| Group size | 5 |
| Max prompt length | 6000 |
| Max response length | 2000 |
| Learning rate | 1e-6 |
| Optimizer | AdamW |
| Weight decay | 0.01 |
| KL coefficient | 0.01 |
| Clip ratio | 0.2 |
| Epochs | 4 |
| Rollout engine | vLLM |
| Temperature | 1.0 |
| Top-p | 1.0 |
| Gradient clipping | Max norm 1.0 |

Table 10: Hyperparameters for RL training for LLM reasoning.

| Hyperparameter | Value |
|---|---|
| Batch size | 16 |
| Optimizer | AdamW |
| Weight decay | 0.0001 |
| Epochs | 4 |
| Learning rate | 1e-6 |
| Scheduler | StepLR |
| Step schedule | [4] |
| Gamma | 0.1 |
| Freeze vision backbone | True |
| Freeze language encoder | True |
| Vision backbone | swin_T_224_1k |
| Language encoder | bert-base-uncased |
| Gradient clipping | Max norm 0.1 |

Table 11: Hyperparameters for GroundingDINO fine-tuning

## A.10 PROMPTS

```
You are an expert at solving spatial reasoning queries. You will be given a query and an API of methods you
can call.

You must produce a step by step plan inside the <plan></plan> tags and then a python program that executes the
 plan inside <answer></answer> tags.

You have the following guidelines:
1. You must produce a text-wise plan inside <plan></plan> tags.
2. Assume each question is asking for 3D measurements, NOT pixel measurements. You must multiply 2D
measurements by depth to get pseudo-3D for comparisons.
3. Answers should NEVER be rounded or ceiling'd - leave all answers as decimals, even when asked for ratios.
4. Height and depth are not the same thing. Height refers to the dimension of the object, namely how tall it
is. Depth refers to the distance of an object from the camera.
5. Be very speciic about which tools to call and how to call them, steps should be very specific.
6. All final answers must be stored in a variable called final_answer in the programs. I will be parsing for
that variable, so you MUST name the output that.
7. You must ensure that the final output step matches the specified problem response type. For example, if the
 question gives some options, the last step must specify that it must return one of the the options.
8. Above, below, under, beneath are 2D relationships and should be handled with x,y coordinates. Behind and in
 front are 3D relationships and should be handled with depth.

Here is your API:
{API}

Here are some examples of how you might solve a problem:
{ICL Examples}

You can assume that the variable img_pth has already been defined. DO NOT override it.
You MUST put a text-based plan inside <plan></plan> tags, and a program inside <answer></answer> tags.
You MUST put your final answer inside a python variable named final_answer in your python program (I will be
extracting this variable).
You MUST NOT round final answers, keep all outputs as decimals (even when you feel it should be rounded).
```

Figure 15: **Program Generation System Prompt**. The API can be found in Fig. 16 and the ICL examples in Fig. 17

```
def depth(img_pth, bbox):
    """
    Estimates the depth of an object specified by a bounding box.

    Parameters
    ----------
    img_pth : str
        Path to the input image file.
    bbox : list[int, int, int, int]
        Object bounding box in form [x1, y1, x2, y2].

    Returns
    -------
    float
        The depth of the object specified by the bounding box.
    """

def gd_detect(img_pth, prompt):
    """
    Run object detection on an image and return the post-processed bounding boxes.
    Not necessarily in the same order as the prompt.

    Parameters
    ----------
    img_pth : str
        Path to the input image file.
    prompt  : str
        Natural-language description of the object to be detected that describes the object's appearance. It
        can be a noun, e.g., "fireplace, coffee table, sofa", or accompanied by an appearance attribute, e.g.,
        "the black coffee table", "the pink pillow".
        NOTE: do not pass spatial relationship attribues, e.g., "the rightmost sofa" is wrong -- they don't
        relate to the object's appearance.

    Returns
    -------
    list[dict]:
        A list where each element is a dict {"bbox": [x1, y1, x2, y2], "label": <str>}. The list is NOT
        necessarily in order of the prompt.
    """

def vqa(img_pth, bbox, prompt):
    """
    Answers a query about an object specified by a bounding box (bbox). The query should relate to the
    appearance of the object and should be simple, e.g. "What color is the object".
    Should not be used for metric values about objects (e.g. object height, depth, width).
    Can be used with bbox None for questions about the image in the event of early exit. In this event, the
    module can be used for broader, wholistic questions.

    Parameters
    ----------
    img_pth : str
        Path to the input image file.
    bbox : list[int, int, int, int] | None
        The bounding box of the object used to ground the visual query in [x1, y1, x2, y2] form.
        Can be set to None for wholistic questions about the image in case of early exit.
    prompt : str
        The question to ask the GPT model. Keep questions simple.

    Returns
    -------
    str
        The response to the query.
    """
```

Figure 16: **API for vision specialist modules**.

```
Question 1: If the 3D height of the two-seat sofa is 0.50 meters, what is the 3D height of the dining table in
 meters?
Solution 1: <plan>
1. Use gd_detect on the image with prompt "two-seat sofa, dining table" to get detections for both objects.
2. Identify which detection corresponds to the sofa (label contains "sofa") and which to the dining table (
label contains "table").
3. If either object is not found, ask vqa the entire query with bbox as None, store this answer in
final_answer and return.
4. For each object:
    a. Extract its bounding box [x1, y1, x2, y2].
    b. Compute the 2D height in pixels as (y2 - y1) and convert to float.
    c. Pass the box to depth to get a scalar depth value for the object.
    e. Multiply the 2D height by the depth to get the pseudo-3D height.
5. Let sofa_pseudo be the pseudo-3D height of the sofa. We are given that its real 3D height is 0.50 meters.
Compute a scale factor = 0.50 / sofa_pseudo.
6. Multiply the dining table's pseudo-3D height by this scale factor to get its real 3D height in meters.
7. Store the result in the variable final_answer without rounding.
</plan></answer>
# Step 1: Detect sofa and dining table
detections = gd_detect(img_pth, "two-seat sofa, dining table")

sofa_det = None
table_det = None
for det in detections:
    label = det["label"].lower()
    if "sofa" in label:
        sofa_det = det
    elif "table" in label:
        table_det = det

if sofa_det is None or table_det is None:
    final_answer = vqa(img_pth, None, "If the 3D height of the two-seat sofa is 0.50 meters, what is the 3D
    height of the dining table in meters?")
    return

# Step 2: Compute pseudo-3D height of the sofa
sx1, sy1, sx2, sy2 = sofa_det["bbox"]
sofa_height_2d = float(sy2 - sy1)
sofa_depth = depth(img_pth, sofa_det["bbox"])
sofa_pseudo = sofa_height_2d * sofa_depth

# Step 3: Compute pseudo-3D height of the dining table
tx1, ty1, tx2, ty2 = table_det["bbox"]
table_height_2d = float(ty2 - ty1)
table_depth = depth(img_pth, table_det["bbox"])
table_pseudo = table_height_2d * table_depth

# Step 4: Convert to real-world meters using the known sofa height
scale = 0.50 / sofa_pseudo
final_answer = table_pseudo * scale
</answer>

Question 2: What is closer to the camera: the tv or the sofa?
Solution 2:<plan>
1. Use gd_detect to detect the TV and sofa in the image, storing their bounding box.
2. If either object is not found, store the other object in final_answer and return.
3. For each detected object (TV and sofa), call depth with the object bounding box to get the depths for each
object (tv_depth, sofa_depth).
4. Compare tv_depth and sofa_depth. If tv_depth < sofa_depth, set final_answer = "tv", else set final_answer =
 "sofa".
</plan><answer>
# Step 1: Detect the TV and sofa
detections = gd_detect(img_pth, "tv, sofa")

sofa_det = None
tv_det = None
for det in detections:
    label = det["label"].lower()
    if "sofa" in label:
        sofa_det = det
    elif "tv" in label:
        tv_det = det

# Step 2: Check detections
if tv_det is None:
    final_answer = "sofa"
    return
if sofa_det is None:
    final_answer = "tv"
    return

tv_bbox = tv_det["bbox"]
sofa_bbox = sofa_det["bbox"]

# Step 3: Query depths
tv_depth = depth(img_pth, tv_bbox)
sofa_depth = depth(img_pth, sofa_bbox)

# Step 4: Compare depths (smaller = closer)
if tv_depth < sofa_depth:
    final_answer = "tv"
else:
```

```
    final_answer = "sofa"
</answer>

Question 3: What is the number of placemats divided by the number of plants?
Solution 3: <plan>
1. Use gd_detect on img_pth with prompt "placemat" to detect all placemats.
2. Use gd_detect on img_pth with prompt "plant" to detect all plants.
3. If either object is not found, ask vqa the entire query with bbox as None, store this answer in
final_answer and return.
4. Count the number of detected placemats (length of placemat detections list).
5. Count the number of detected plants (length of plant detections list).
6. Compute the ratio: number_of_placemats divided by number_of_plants.
7. Store the result in the variable final_answer.
</plan><answer>
# Step 1: Detect all placemats
placemat_dets = gd_detect(img_pth, "placemat")

# Step 2: Detect all plants
plant_dets = gd_detect(img_pth, "plant")

# Step 3: Check detections
if len(placemat_dets) == 0 ot len(plant_dets) == 0:
    final_answer = vqa(img_pth, None, "What is the number of placemats divided by the number of plants?")
    return

# Step 4: Count placemats
num_placemats = len(placemat_dets)

# Step 5: Count plants
num_plants = len(plant_dets)

# Step 6: Compute the ratio
final_answer = num_placemats / num_plants
</answer>

Question 4: Which object is closer to the camera: the washing machine or the rightmost chair? Options: {
washing machine, rightmost chair}
Solution 4: <plan>
1. Use gd_detect on img_pth with prompt "washing machine" to detect the washing machine.
2. Use gd_detect on img_pth with prompt "chair" to detect all chairs.
3. If either object is not found, store the other object in final_answer and return.
4. For each chair detection, compute its 2D center x-coordinate as (x1 + x2) / 2; select the detection with
the maximum center x as the rightmost chair.
5. For both the washing machine and the rightmost chair, call depth with the object bounding box to get depths
 for each object (wm_depth, chair_depth).
6. Compare wm_depth and chair_depth: the smaller depth value indicates the closer object.
8. If wm_depth < chair_depth, set final_answer = "washing machine"; otherwise set final_answer = "rightmost
chair".
</plan><answer>
# Step 1: Detect the washing machine
wm_dets = gd_detect(img_pth, "washing machine")

# Step 2: Detect all chairs
chair_dets = gd_detect(img_pth, "chair")

# Step 3: Check detections
if len(wm_dets) == 0:
    final_answer = "rightmost chair"
    return
if len(chair_dets) == 0:
    final_answer = "washing machine"
    return

# Step 4: Identify the rightmost chair by max center x
def center_x(det):
    x1, y1, x2, y2 = det["bbox"]
    return (x1 + x2) / 2.0

rightmost_chair_det = max(chair_dets, key=center_x)
wm_det = wm_dets[0]

# Step 5: Query depths for both objects
wm_depth = depth(img_pth, wm_det["bbox"])
chair_depth = depth(img_pth, rightmost_chair_det["bbox"])

# Step 6 & 7: Compare depths (smaller = closer)
if wm_depth < chair_depth:
    final_answer = "washing machine"
else:
    final_answer = "rightmost chair"
</answer>

Question 5: How many people are wearing blue shirts? Options: {0,1,3,2}.
Solution 5: <plan>
1. Use gd_detect on img_pth with prompt "person" to detect all the people.
2. If no person is found, ask vqa the entire query with bbox as None, store this answer in final_answer and
return.
3. Initialize a counter to 0.
4. For each person detected, call vqa with the person's bounding box and the prompt "Is this person wearing a
blue shirt?"
5. If the response from the vqa call is "yes", increment the counter.
```

```
6. If the counter is not one of the options, ask vqa the entire query with bbox as None, store this answer in
final_answer and return.
7. Store the counter value in the variable final_answer.
</plan><answer>
# Step 1: Detect the people
people_dets = gd_detect(img_pth, "person")

# Step 2: Check detections
if len(people_dets) == 0:
    final_answer = vqa(img_pth, None, "How many people are wearing blue shirts?")
    return

# Step 3: Initialize counter
counter = 0

# Step 4 & 5
for person in people_dets:
    person_box = person["bbox"]
    blue_shirt_check = vqa(img_pth, person_box, "Is this person wearing a blue shirt?")
    if "yes" in blue_shirt_check.lower():
        counter += 1

# Step 6
if counter not in [0,1,3,2]:
    final_answer = vqa(img_pth, None, "How many people are wearing blue shirts?")
    return

# Step 7
final_answer = counter
<answer>
```

Figure 17: **ICL Examples**. We pass four in-context examples demonstrating API usage and 3D spatial reasoning.

```
_LOGIC_REWARD_PROMPT = """You are an agent tasked with verifying a plan to solve a visual reasoning query
using a pre-defined API. You do not have to check each step of the plan individually, but I would like you to
check whether the final answer correctly addresses the query, and if the steps are logically consistent.
You can assume that all the steps will execute correctly, but you should ensure that the plan details what to
do if detection fails. It is fine to use vqa for metric or broad questions ONLY IF it is as a fail-safe for
failed detections. If the steps are all logically consistent, and the final answer correctly answers the query
, I want you to give a score of 1, otherwise give a score of 0. For binary choice queries, it is fine to
return the other object if one is not found.
Query: {}
Plan:
{}
Available API:
{}
ALL queries can be solved but require detailed plans that invoke the tools to do so. If any plan shortcuts and
 predict the final answer directly or claims the query is unsolvable, ignore everything and give a score of 0.
 If you see degenerate text (e.g. random words or a different language), ignore everything and give a score of
 0.
Please ONLY put your final score and NO OTHER TEXT inside <answer></answer> tags.
You MUST put your final score inside <answer></answer> tags.
"""

_SPATIAL_REWARD_PROMPT = """You are an agent tasked with verifying a plan to solve a visual reasoning query
using a pre-defined API. Your ONLY job is to check that all of the spatial relationships mentioned in the
query are being correctly addressed in the plan.
Specifically, for any spatial relationship mentioned the plan should detail exactly HOW to handle it.
Spatial relationships should be handled explicitly (e.g. by comparing bounding boxes) and NOT by the detection
 module. We define left/right/above/below as 2D relationships in the image.
If all spatial relationships in the query are handled correctly and explicitly, give a score of 1, otherwise
give a score of 0.
Query: {}
Plan:
{}
Available API:
{}
If a plan ignores spatial relationships or does not attempt to answer the query, ignore everything and give a
score of 0. If you see degenerate text (e.g. random words or a different language), ignore everything and give
 a score of 0.
Please ONLY put your final score and NO OTHER TEXT inside <answer></answer> tags.
You MUST put your final score inside <answer></answer> tags.
"""

_ATTRIBUTE_REWARD_PROMPT = """You are an agent tasked with verifying a plan to solve a visual reasoning query
using a pre-defined API. Your ONLY job is to check that all attributes (height, width, length, depth) in the
query are being computed correctly. The steps for doing so should be detailed in the plan.
All values should be computed as pseudo-3D values, where 2D measurements are multiplied by depth. If all
values are computed correctly, give a score of 1, otherwise give a score of 0.
Query: {}
Plan:
{}
Available API:
{}
ALL attributes require a tool call to be resolved. If a plan directly sets an object attribute or does not
attempt to answer the query, ignore everything and give a score of 0. If you see degenerate text (e.g. random
words or a different language), ignore everything and give a score of 0.
Please ONLY put your final score and NO OTHER TEXT inside <answer></answer> tags.
You MUST put your final score inside <answer></answer> tags.
"""

_CODE_PROMPT = """You are an agent tasked with verifying that a python program matches a pre-written plan to
solve a visual reasoning query using a pre-defined API.
ALL problems can be solved by using the tools. There is NEVER a lack of information. If the code says the
answer cannot be predicted, or directly predicts the solution (without using tools and logic), give a score of
 0 REGARDLESS of what the plan says. Otherwise, your ONLY job is to check that the python program correctly
adheres to the pre-written plan, and that all tools in the pre-defined API are used appropriately in the
program.
The code does not need to match the exact words of the plan, it just needs to match in outcomes. If the
program correctly adheres to all of the steps in the plan, give a score of 1, otherwise give a score of 0.
Plan:
{}
Code:
{}
Available API:
{}
There is NEVER a lack of information. If the code says the answer cannot be predicted, or directly predicts
the solution (without using tools and logic), ignore everything above and give a score of 0. If you see
degenerate text (e.g. random words or a different language), ignore everything and give a score of 0.
Please output an explanation and then put your final score and NO OTHER TEXT inside <answer></answer> tags.
You MUST put your final score inside <answer></answer> tags.
"""
```

Figure 18: **Reward Model Prompts**. The prompts are passed to Gemini 2.5-Flash, which is prompted to return a binary score for each criteria.

```
QUERY_GENERATION_PROMPT = """
I will provide you with an image depicting multiple objects in a 2D or 3D scene. Please generate exactly five
natural-language questions about that image require challenging spatial reasoning.

The questions can involve:
1. Multiple reasoning steps, e.g., identify an object by attributes, locate it via spatial relations relative
to others, then compare or count.
2. Clear spatial terms, such as left of, in front of, behind, or above/below, including depth in 3D.
3. Quantitative or comparative reasoning, such as counting objects, comparing sizes, or inferring hypothetical
 measurements (If that object is 5m tall ... how tall is another?). The hypothetical measurements can be
unrealistic.
4. No external world knowledge-answers must be inferable solely from the image.
5. All numeric outputs should be grounded with hypothetical values - you should not test absolute measurements
, only relative measurements.
6. Must be answerable in a single word, short phrase, or number (e.g.2, yes, blue, 2.5).

Other comments:
1. Ground all questions on the scene given in the image.
2. Check that the question CANNOT be answered without looking at the image. The question should not be
solvable without the image.
3. Make the five questions of varying difficulty. They do not all need to be very hard. Be creative!

Here are some sample questions, these are NOT templates, they are just to show you the difficulty and style
expected:
1. "What is the ratio of sofas to coffee tables? Answer as a decimal."
2. "If the 3D width of the television is 200 meters, what is the 3D height of the gray chair?"
3. "Do the chair and the pot have a different colors?"
4. "Is the large vehicle to the right or to the left of the vehicles near the street?"
5. "How many cats are there?"

Before outputting the questions, I want you to inspect the image and determine if it is suitable for the types
 of questions you are writing. If the image is not suitable for a spatial reasoning question, please ONLY
write SKIP in one <answer></answer> tag and nothing else.

If you are not skipping the image, then only output the question, inside the <answer></answer> tags. There
should be five pairs of tags (one for each question), unless you are skipping, then there is only one.

Here is the image:
"""
```

Figure 19: Prompt for generating spatial reasoning queries using Gemini-2.5-Flash.

```
GPT_5_PROMPT = """
You are responsible for verifying answers. I will give you a predicted answer and the ground truth answer.
The two may not always be an exact match, your job is to check if they are semantically similar.
You should return a score of 1 if they are semantically roughly the same or 0 if they are not.
Please put your answer inside <answer></answer> tags.
Question: {}
Ground Truth: {}
Predicted Answer: {}
"""
```

Figure 20: **GPT-5-mini synonym equivalence prompt.** We use GPT-5-mini as a judge for the GQA benchmark to account for synonyms in the open-ended answers of the benchmark.

```
COARSE_FILTERING_PROMPT = """
You are an Object-Detection Verifier.
You will receive:
- CLEAN image (truth)
- ANNOTATED image (same image with numbered boxes + labels). The number corresponds to the box that it is
INSIDE OF.
- A numbered list of detections (index, label, [x1,y1,x2,y2])
Your task:
Decide which detections to KEEP and which to DROP.
Return ONLY the JSON required by the schema: {"keep_indices":[...], "drop_indices":[...], "notes":[...]}.
Definitions (judge against the CLEAN image):
- Correct: the dominant object of the box is an instance of the labeled object; label matches; box is
reasonably tight (some background or a small part of another object visible in the borders of the box is OK).
- Incorrect: drop for any of:
  - wrong_label---contents don't match the label or no clear object.
  - bad_box---far too loose/tight, mostly background, contains the entirety of multiple instances, or chops
  off the object.
Indexing:
- Indices are 1-based and must be fully partitioned into keep or drop. No omissions or repeats.
- keep_indices should be sorted by input order.
Notes (short, per decision):
- "4 wrong_label (label=cat, object=dog)"
- "5 bad_box (cuts off left side)"
Never propose new boxes or relabel. When uncertain, favor precision (drop).
Return ONLY the JSON object-no prose.
"""

PER_CROP_PROMPT = """
"""You are a single-box verifier.
Input:
- One CROPPED image.
- One predicted label
Goal:
Decide whether the **main visible object in the crop clearly matches the label**.
Rules (be strict):
- Set keep=true only if the labeled object is the **dominant subject**, is **clearly visible**, and is **
recognizably** that class. The correct object should dominate the image.
- Minor background is fine; the object should not be tiny or incidental. It is acceptable if a small part of
another object is visible in the borders of the image -- but it should not take up more than 20 percent of the
 image.
Set keep=false for any of the following:
- Wrong class: the visible object is a different category, or no clear object of the labeled class is present.
- Truncation: the crop cuts off a substantial part so the class cannot be confidently verified.
- Unclear central object: crop includes several objects and it's not clear which one matches the label.
- Multiple objects: crop includes significant portions/all of several objects of the correct class, not just
one.
Output:
Return ONLY JSON matching the caller's schema, e.g.:
{"keep": true/false, "reason": "very brief justification"}.
No extra text or markdown.
"""

DEDUPLICATION_PROMPT = """
You are the FINAL-PASS object-detection verifier.
You will receive:
- CLEAN image (truth)
- ANNOTATED image (same image with numbered boxes + labels). The number corresponds to the box that it is
INSIDE OF.
- A numbered list of detections (index, label, [x1,y1,x2,y2])
Goal:
Do a last sanity sweep on the remaining detections. Remove any **incorrect** boxes and resolve **residual
duplicates**-with special attention to nested ('sub-object') boxes and boxes that are very close to each other
.
What to KEEP (correct):
- The box contains one main, dominant object.
- Label matches the object class.
- Box is reasonably tight without chopping the object.
- It is acceptable if a small part of another object is visible in the borders of the box -- but it should not
 take up more than 20 percent of the box.
DROP if any of these apply:
- **wrong_label**---contents don't match the label, or no clear object of that class.
- **duplicate**---another detection refers to the same physical instance (rules below).
Duplicate rules (same/compatible class):
Treat two detections as duplicates of the **same** instance if ANY is true:
1) They substantially overlap and the dominant object of both boxes is the same.
2) One box lies almost entirely inside the other (nested/sub-object case).
3) Their centers and extents are almost the same (nearly coincident), even if overlap isn't huge.
Resolving duplicates:
- Keep **exactly one** per duplicate group; drop the rest.
- Prefer the box that best covers the full object with the least extra background.
- If the tighter box chops the object, keep the larger.
- If still unsure, keep the earlier index.
Indexing:
- Indices are 1-based and must be fully partitioned into keep or drop. No omissions or repeats.
- keep_indices should follow input order.
Notes (short, per decision):
- "7 duplicate_of 3 (nested; 3 covers full object)"
- "12 duplicate_of 9 (nearly coincident; kept tighter 12)"
- "4 wrong_label (label=fireplace; object=tv)"
- "5 bad_box (too loose; mostly background)"
Output:
Return ONLY the JSON required by the schema: {"keep_indices":[...], "drop_indices":[...], "notes":[...]}.
```

```
No prose or markdown.
"""
```

Figure 21: **VLM Verifier Prompts**. We use GPT-5-mini as our VLM verifier in a three-step verification process.