# OpenReview forum: "No Labels, No Problem: Training Visual Reasoners with Multimodal Verifiers"
_ICLR.cc/2026/Conference — ICLR 2026 Poster_

### Official Review · Reviewer_5PmE · 2025-10-23

**Soundness:** 2
**Presentation:** 1
**Contribution:** 2
**Rating:** 2
**Confidence:** 4

**Summary:**

This work applies multimodal-based verifiers in form of tool-use in order to improve the visual reasoning capabilities of a base model. These verifiers are used to finetune the systems program and visual grounding modules. The final approach is evaluated on VQA tasks and compared to monolithic as well as program synthesis based models.

**Strengths:**

The experimental results across a diverse set of datasets appear promising.

**Weaknesses:**

General Comments:
I find the paper's overall narrative and key contributions difficult to follow. Below, I outline specific areas that require clarification.

Conceptual Clarity:
	1.	Method characterization: The authors describe their approach as "label-free finetuning via pretrained model verifiers." Could the authors clarify how this fundamentally differs from knowledge distillation from the verifiers into the base model(s)? A clearer distinction would help readers understand the novelty of the approach.
	2.	Code execution mechanism: Section/Figure 2 shows images being passed to an executor, but the technical details of how Python code is actually executed over the image remain unclear. Specifically, what representation is used for this? The current presentation makes this process appear overly simplistic or unclear.
	3.	Vision specialist integration: The paper would benefit from a clearer explanation of how vision specialist models are integrated into the overall system architecture.

Presentation Issues:
Figure readability: Most figures contain text boxes that are too small to read, severely limiting their utility. If the information in these figures is essential for understanding the authors' arguments and results, they must be made readable. Currently, I cannot verify the claims or follow the clarifications the authors make in reference to these figures.

Experimental Design:
	1.	Evaluation objectives: For several experiments, the specific goal and relevance to the main claims are unclear. I recommend adding brief statements at the beginning of each evaluation section explicitly stating what research question is being addressed.
	2.	VALOR vs. VALOR (RL): The distinction between these two variants is not clearly explained in the text. Please provide a concise description of the key differences.
	3.	Verifier and base model specification: Critical information about which models serve as verifiers versus base models appears to be missing (or I may have overlooked it). This should be clearly indicated in the results tables alongside the VALOR specifications, as the choice of verifier likely has substantial impact on performance.

Critical Concern - Experimental Comparisons:
I find the experimental comparisons difficult to interpret due to potentially confounded variables. For example, when comparing VALOR with Qwen3-8B:
	•	Was Qwen3-8B also finetuned in any way?
	•	Was Qwen3-8B used as a verifier in VALOR's training?
	•	The approaches appear fundamentally different, with VALOR receiving substantially more information, possibly larger models as verifiers and additional training. Doesn't this alone explain the performance differences?
Without clarity on these points, it's difficult to assess whether the comparisons are fair or what specific conclusions can be drawn.

Recommendations:
	1.	Ablation study on verifiers: Given the central role of verifiers in the method, an ablation study examining how different verifiers affect performance would be valuable. This would help isolate the contribution of the VALOR framework from the contribution of powerful verifiers.
	2.	Clarify contributions: Each evaluation section would benefit from a concluding paragraph that: (a) summarizes the key findings, (b) relates them back to the paper's main claims, and (c) explains their significance for the overall contribution.
	3.	Sharpen claims: Throughout the paper, the goals and contributions could be stated more crisply and specifically, making it easier for readers to follow the logical thread from motivation through methods to conclusions.

Overall Assessment:
Due to the issues outlined above, I find it challenging to fully evaluate the validity of the authors' claims. Addressing these concerns would significantly strengthen the paper.

**Questions:**

It seems VALOR-RL performs worse than VALOR in nearly all evaluations. Whats the intuition behind this? Is this just an error in the training set generation?

VALOR vs proprietary models: I'm a little confused what the benefit is of comparing to different propietary models, couldn't VALOR be applied on top of these as well?

Why were different dataset subsets used across the different evaluations?

What is the difference between evaluations in section 4.4 and 4.1?

---

> ### Author Response · Authors · 2025-11-20
> **Response to 5PmE (part 1)**
>
> We thank you for taking the time to write your review. We address your comments and concerns below. Please let us know if you have any additional questions.
>
> > The authors describe their approach as "label-free finetuning via pretrained model verifiers." Could the authors clarify how this fundamentally differs from knowledge distillation from the verifiers into the base model(s)?
>
> We believe there are key fundamental differences. Knowledge distillation typically transfers *output predictions* or *model weights* from a teacher into a student model. In contrast, our method uses pretrained models as *verifiers* that provide *binary correctness* *feedback* over structured intermediate outputs (plans, executable code, grounding detections), rather than supervising the final answer. This distinction leads to concrete differences:
>
> 1\. **Our learning signal is reinforcement-based, not distillation based.** Verifiers do not provide target responses, they simply evaluate the model outputs based on pre-defined criteria. Thus, our model *discovers* improved policies through RL \- not distillation.
> 2\. **Our model is not trained to mimic the verifier’s answer distribution.** By only providing a coarse judgement of correctness as opposed to full output solutions, our trained model is not bound by the answer distribution of the verifier (as is the case in distillation).
>
> > Section/Figure 2 shows images being passed to an executor, but the technical details of how Python code is actually executed over the image remain unclear. \[...\] The paper would benefit from a clearer explanation of how vision specialist models are integrated into the overall system architecture.
>
> Given a query, the LLM generates a natural language plan followed by a corresponding program in Python **(L130-131)**. The Python program can call three pre-defined functions that invoke the vision specialist models **(L136-137)**. The vision specialists are all open-source models that can be loaded and executed in Python. This process is described extensively in the main paper **(L130-131, 135-138, 160-161, 289-293)**, calls to the specialist models are shown in **Figure 1** (gd\_detect, depth, vqa), **Figure 2**, **Figure 3**, and **Figure 4**. In the Appendix, these are again shown in **Figure 8**. Moreover, additional details on the vision specialists are included in **A.2**, and all prompts for code-generation and invoking the vision specialists are in **Figures 13-15**. We note that reviewers rpw5, 3T5W, DLBC have all characterized our approach accurately in their summaries. However, if you feel there are clarifying details we have omitted, please let us know and we can update the manuscript.
>
> > Most figures contain text boxes that are too small to read, severely limiting their utility. If the information in these figures is essential for understanding the authors' arguments and results, they must be made readable. Currently, I cannot verify the claims or follow the clarifications the authors make in reference to these figures.
>
> Our model produces Python code as output. We cannot provide qualitative examples, which we believe are crucial for highlighting model behavior, without including programs. We have ensured the programs are included as PDFs, so they are entirely visible and legible when zooming in – which we recommend in the figure caption (**L355**). We additionally describe the figure contents extensively in the manuscript (**L315-319, 369-373, 408-412**). If you have suggestions for how to include Python programs in a more reader-friendly format, we would be happy to incorporate your feedback into our figures.

---

> ### Author Response · Authors · 2025-11-20
> **Response to 5PmE (part 2)**
>
> > For several experiments, the specific goal and relevance to the main claims are unclear. I recommend adding brief statements at the beginning of each evaluation section explicitly stating what research question is being addressed
>
> We detail the goal and evaluation details of each experimental comparison when introducing the experiments (**L292-293, 294-295, 297-298, 299-300**) and additionally in the first paragraph of each experimental subsection (**L304-308, L377-404, L423-430, L437-444**). Furthermore, we have sentences contextualizing experimental findings throughout the subsequent paragraphs (**L316-319, 320-322, 359-360, 366-367, 408-409, 413-416, 431-433, 449-450)**. However, for your convenience, we re-summarize the goal of our experiments below:
>
> 1. (4.1) LLMs with tools: we prompt proprietary and open-source LLMs (language-only) with a query to generate plans and programs using our vision-specialist APIs. We then execute these programs and report accuracy. The goal of this experiment is to (a) quantify the impact of verifier-improved reasoning (VALOR-RL vs Qwen3-8B; **L309-311**) and (b) quantify the impact of verifier-improved visual grounding (VALOR VS VALOR-RL; **L320**).
> 2. (4.2) Visually grounded, RL-tuned VLMs: we compare VALOR to GRIT and ViGoRL, which both fine-tune VLMs for visual grounding tasks but rely on ground truth labels for training. The goal of this experiment is to contrast our label-free approach against RL-tuned methods with labels (**L294-295**). Additionally, this experiment highlights the difference between reasoning in text without tools (as GRIT and ViGoRL do), and explicitly grounding with tools (**L413**).
> 3. (4.3) Visual program synthesis methods: we compare VALOR against other methods that produce executable programs using LLMs (VADAR, ViperGPT, VisProg). This experiment contrasts our proposed annotation-free training framework with training-free program synthesis methods that use proprietary LLMs for inference (**L296-297, 426-428, 433-444**).
> 4. (4.4) Direct-answer VLMs: we prompt proprietary and open-source VLMs to directly predict answers from (image, query) pairs. This experiment contrasts direct VLM prediction with our two-step reasoning and tool use paradigm (**L299-300**).
>
> > When comparing VALOR with Qwen3-8B: • Was Qwen3-8B also finetuned in any way? • Was Qwen3-8B used as a verifier in VALOR's training? The approaches appear fundamentally different, with VALOR receiving substantially more information, possibly larger models as verifiers and additional training. Doesn't this alone explain the performance differences? Without clarity on these points, it's difficult to assess whether the comparisons are fair or what specific conclusions can be drawn.
>
> We feel this comment stems from a fundamental misunderstanding of our work. We are not introducing another competing LLM, we are providing a recipe to **tune and improve existing LLMs without ground truth supervision** (**L15-20, 66-68, 129-134**). This contribution was well-characterized by **Reviewer DLBC** “\[...\] propose an annotation-free **training framework**”, **Reviewer 3T5W** “\[...\] **improve** spatial and reasoning-heavy visual understanding tasks without human-annotated labels.”, and **Reviewer rpw5** “\[...\] a novel **visual reasoning training paradigm**, VALOR, which enables models to improve both reasoning and visual grounding capabilities without any manual annotations.” Our goal in comparing VALOR-RL with Qwen3-8B is to quantify the impact of verifier-improved reasoning (**L309**).
>
> Regarding comparisons to Qwen3-8B, we compare VALOR-RL directly against the pre-trained Qwen3-8B model. We describe the Qwen3-8B model as a “**base model**” (**L311**), implying it has not been post-trained in any way. We are happy to write this more explicitly if you feel this is not sufficiently implied. We use Gemini-2.5-Flash as the LLM verifier used to train VALOR-RL (**L956**), not Qwen3-8B. As outlined in the text (**L309**), the goal of this comparison is to isolate the impact of verifier-improved reasoning. **VALOR-RL** uses a verifier-trained Qwen3-8B model (**L311)** with the same visual grounding modules (**L307-308**). Thus, improvements from Qwen3-8B to VALOR-RL stem from our first stage of training (§3.1).

---

> ### Author Response · Authors · 2025-11-20
> **Response to 5PmE (part 3)**
>
> > VALOR vs. VALOR (RL): The distinction between these two variants is not clearly explained in the text.
>
> The distinction between these two models is that **VALOR** uses the verifier-improved visual grounding model (§3.2), while **VALOR-RL** only features verifier-improved reasoning (§3.1). This is described explicitly in the title of the paragraph comparing the two variants “VALOR vs VALOR-RL: **The** **impact of verifier-improved visual grounding**” (**L320**), as well as the introduction of VALOR-RL in §3.1 (**L215**; prior to §3.2 which introduces the verifier-improved visual grounding module present in VALOR). We further re-iterate this difference in the same paragraph comparing the two variants : “VALOR and VALOR-RL share programs, isolating contributions from our improved visual grounding model (§3.2)” (**L321-323**).
>
> > Verifier and base model specification: Critical information about which models serve as verifiers versus base models appears to be missing (or I may have overlooked it)
>
> We include details of the verifiers in the Appendix (**L856-857, 956**). We use Gemini 2.5-Flash as our LLM verifier and GPT-5-mini as our VLM verifier. We use Qwen3-8B as the base model for VALOR (**L213**, **311**).
>
> > Ablation study on verifiers: Given the central role of verifiers in the method, an ablation study examining how different verifiers affect performance would be valuable.
> Thank you for the helpful suggestion. We agree that examining how different verifiers affect performance would provide valuable insight. Conceptually, VALOR is verifier-agnostic – any model capable of assessing spatial reasoning correctness can serve as the verifier – and we selected Gemini-2.5-Flash (L856–857) because it was the strongest model available to us at the time.
>
> We sample 100 model outputs during training and manually compute the reward. We consider this set of manually annotated outputs as the ground truth. To establish baseline performance, on the same 100 outputs, we find that our LLM verifier (Gemini-2.5-Flash) matches our ground truth $87\\%$ of the time. On the samples that do not match, Gemini tends to under-reward compared to our ground truth.
>
> We then repeated the same evaluation using two smaller open-source models – Qwen3-8B and Llama-3.2-11B-Instruct – as verifiers. Their reward predictions matched the ground truth $15\\%$ and $7\\%$, respectively. These results reaffirm our choice of Gemini-2.5-Flash for the main experiments and highlight the current gap between large proprietary models and smaller open-source models in their ability to reliably verify spatial-reasoning programs.
>
> We will add this analysis to the revised manuscript.
>
> > It seems VALOR-RL performs worse than VALOR in nearly all evaluations. Whats the intuition behind this?
>
> VALOR executes the same programs as VALOR-RL, but uses the verifier-improved visual grounding module (**L320-322**). VALOR outperforming VALOR-RL shows that our verifier-based training (§3.2) is successfully improving the detection module (**L320**).

---

> ### Author Response · Authors · 2025-11-20
> **Response to 5PmE (part 4)**
>
> > VALOR vs proprietary models: I'm a little confused what the benefit is of comparing to different propietary models, couldn't VALOR be applied on top of these as well?
>
> No. VALOR is an annotation-free training paradigm (**L67**). We cannot train proprietary models as they are closed-weights.
>
> > Why were different dataset subsets used across the different evaluations?
>
> For Table 3 we report on Omni3D-Bench and GQA as those are the benchmarks reported for VisProg, ViperGPT, and VADAR in the VADAR paper. We do this in an attempt to be more favorable to those methods, as they were likely designed for the benchmarks they report. In Table 4, we only consider benchmarks released after the release dates of the VLMs to ensure these benchmarks are not present in the training data of the VLMs (**L439-444**).
>
> > What is the difference between evaluations in section 4.4 and 4.1?
>
> In the evaluations in Section 4.1, LLMs are prompted “with a query to generate plans and programs using our vision-specialist APIs”. We then execute those programs and determine accuracy. The goal of Section 4.1 is to evaluate the improvement to LLM reasoning with verifiers (**VALOR-RL vs open-source LLMs; L309)**, and further the impact of verifier-improved visual grounding (**VALOR vs VALOR-RL; L320)**. In Section 4.4, we report accuracy of VLMs asked to predict the answer directly from (image, query) pairs (**L438-439**). The goal of Section 4.4 is to contrast VLM prediction with our reasoning and tool use paradigm (**L300**).  We note that we explicitly specify these differences at the introduction of our experiments (**L289-300**), the start of Section 4.1 (**L304-308**), and the start of Section 4.4 (**L437-444)**.
>
> We hope our comments have addressed sources of confusion. If there are any remaining questions or comments we would be happy to address them.

---

> ### Author Response · Authors · 2025-11-25
> **Follow-up response to 5PmE**
>
> Hi Reviewer 5PmE, we thank you again for the time taken to write your review. We wanted to follow up to ask if you had any remaining questions or concerns. Thank you.

---

> > ### Comment · Reviewer_5PmE · 2025-11-27
> >
> > Thanks for the response.
> >
> > Ok thanks, I understand the distinction between distillation and verifier based RL.
> >
> > I think one thing that confused me upon first evaluation is what the underlying task is for the model. Currently it's described quite vaguely, in that there is a query and the model generates code for the query. But I think it would help me to specify the underlying task/domain of these queries. So are these visual question answering type of queries, etc.? This would make it easier to understand what the program is generated for. In other words, the authors have responded with "Given a query, the LLM generates a natural language plan ..." My question is a plan for what? This task setting isn't well defined in the problem setup in section 3.
> >
> > Ok I understand now that the vision models are callable functions which can be integrated into code. Again I think a drawback is that the images aren't giving much of a benefit for me. Either they contain very abstract boxes or very small (provocatively: unreadable) code. Maybe try to break the code down to core aspects it should be showing? I.e. the main figures do not need to show the actual code that your approach is outputting, this is just a schematic. It is more beneficial in my view to have an easy to understand overview of the main aspects of some of these figures and put the original code in the appendix for interested readers or other plots in the main paper. But having to read line by line a full programming can be somewhat overwhelming. In this context I think it would really help to have a running example based on a more simple program in one of the figures (figure 1 or 2 potentially) to understand each aspect of the approach, but for this requires that each aspect is well readable.
> >
> > Regarding unclear experiments. Thanks for the summary for me. I think I am missing what the Valor setup is. I.e., why are the authors comparing VALOR to three different models. I would expect something more like: Qwen, Qwen+VALOR, Qwen+VALOR+RL, Gemma, Gemma+valor, Gemma+valor+RL, etc.. I.e. I can't find what the experimental instantiation is of "VALOR" (what base model?) and then I am wondering if we are comparing apples with oranges here, if VALOR is e.g. based on a very different model. In other words VALOR seems to be a training setup which you can apply to many different models. And this issue to me is in evaluation 4.1, 4.2 and 4.3. E.g. in Table 2 the authors have used Qwen3 and compared to GRIT with Qwen 2.5. I understand by the numbers that is unlikely that the difference in performance is only in the base model version, but I think its difficult to fully grasp the potential advantage of VALOR without having the same base model.
> > I see now that the authors have specified the use of Qwen 3 as base model in line 311. This definitely needs to be more prominent. I suggest a dedicated section on experimental setup of VALOR. The authors detail the datasets and baselines, but not VALOR itself. This should include things that are in the appendix as the authors state about the verifier details (using Gpt-5 etc) .
> >
> > Proprietary models: certainly they have closed weights, but in principle if you had access to these you could apply VALOR just as well on them. This is just something I think the authors could/should note in the text.
> >
> > Evaluation 4.4: I still don't understand the difference to evaluation 4.1. Why am I seeing different numbers, e.g., for Llama-3.2-11B on the OMNI3d dataset between table 4 and table 1?
> >
> > I hope my comments are clear, let me know otherwise.

---

> > > ### Author Response · Authors · 2025-11-30
> > > **Follow-up response to 5PmE (part 1)**
> > >
> > > We thank you for taking the time to read our response. We are happy to have resolved some sources of confusion. We address your remaining comments and questions below.
> > >
> > > > I think it would help me to specify the underlying task/domain of these queries. So are these visual question answering type of queries, etc.? In other words, the authors have responded with "Given a query, the LLM generates a natural language plan ..." My question is a plan for what? This task setting isn't well defined in the problem setup in section 3\.
> > >
> > > Our work tackles the task of visual and spatial reasoning. We describe the task in the abstract (**L11-12**, **22-23**), Introduction (**L29-31, 42-43, 73-75**), Method section (**L129-131, 179-180, 201, 203-205, 208, 209-212, 246**), and Experiments section (**L259-260, 264-265**). Figure 1 (the teaser figure of our work) provides a qualitative example of the task, with additional examples shown in Figure 4\.
> > >
> > > We note that the other three reviewers have done an excellent job summarizing our work, which you might also find helpful. Specifically, reviewer DLBC correctly characterized our work as addressing the “problem of query-based visual reasoning”, reviewer 3T5W notes that VALOR “achieve\[s\] significant gains in compositional and spatial visual reasoning”, and reviewer rpw5 writes that “VALOR surpasses both open-source and proprietary models across a broad range of spatial reasoning benchmarks.”
> > >
> > > We are happy to add “spatial reasoning” to the sentence you have highlighted (“Given a query …”) if you feel we have not been sufficiently explicit in Section 3\.
> > >
> > > > The main figures do not need to show the actual code that your approach is outputting, this is just a schematic. It is more beneficial in my view to have an easy to understand overview of the main aspects of some of these figures and put the original code in the appendix for interested readers or other plots in the main paper
> > >
> > > We agree that the code in the figures requires zoom. We note that they are PDFs and thus legible under some zoom. We feel it is imperative to include code in the figures as it would be challenging to highlight model behavior without including it. Including model outputs and figures with code is standard practice for visual programming methods. Additionally, for Figure 3, we feel the model output is necessary to highlight the role of each reward head. However, following both your suggestion and that of reviewer DLBC, we are happy to adjust the presentation of Figure 4 to attempt to make the code larger and show only trimmed versions of the model output.
> > >
> > > > In other words VALOR seems to be a training setup which you can apply to many different models
> > >
> > > Yes you are correct that VALOR is a training framework. We chose Qwen3 as our base model as it was the best open-weight base model (**L310-311**). Our goal was to show that there are significant gains to be made when training with VALOR when tuning an already competent model. We note that by doing so, VALOR performs favorably when compared to proprietary LLMs despite being significantly smaller (**L364-366**). Regarding comparisons, we note that throughout all experiments, all variants of VALOR use Qwen3 as the base model (**L213, 311**), enabling a direct comparison with Qwen3. To allow further comparisons on the impact of verifier-improved grounding, we report Gemma3 and Qwen3 with the verifier-improved visual grounding model in our response to Reviewer rpw5, which we will add to our camera-ready manuscript.
> > >
> > > > In Table 2 the authors have used Qwen3 and compared to GRIT with Qwen 2.5
> > >
> > > You are correct that GRIT and ViGoRL use Qwen2.5-VL, whereas our method uses Qwen3 as the base model. We clarify the purpose of these comparisons:
> > >
> > > The approaches are fundamentally different: GRIT and ViGoRL rely on a VLM to directly perform visual grounding and reasoning jointly inside the VLM (**L377**). In VALOR, however, the base model’s role is entirely different: we require a **language-only** LLM whose purpose is to generate executable programs to solve a spatial reasoning query, not to perform visual perception. Visual grounding is delegated to external specialist modules. Despite the methods differing critically in their use of large models (VLM vs LLM as base), we believe that the comparison of methods using the same family of models (Qwen series) is still valuable. The comparison with these methods (both quantitative and qualitative) is to show how they differ in principle for both grounding (in-text without tools vs tool-use via detectors) and reasoning. We describe these differences extensively in the analysis of Table 2 and Figure 4 (**L406-412**) and explicitly address grounding in the discussion of Figure 5 (**L413-419**).

---

> > > > ### Author Response · Authors · 2025-11-30
> > > > **Follow-up response to 5PmE (part 2)**
> > > >
> > > > > Proprietary models: certainly they have closed weights, but in principle if you had access to these you could apply VALOR just as well on them.
> > > >
> > > > Yes, certainly if we had access to the weights of proprietary models we could apply the VALOR training framework.
> > > >
> > > > > Evaluation 4.4: I still don't understand the difference to evaluation 4.1. Why am I seeing different numbers, e.g., for Llama-3.2-11B on the OMNI3d dataset between table 4 and table 1?
> > > >
> > > > As written in our previous response, in Section 4.1, LLMs are prompted “with a query to generate plans and programs using our vision-specialist APIs”. We then execute those programs and determine accuracy. In Section 4.1, all base models are prompted with **language-only** and produce Python programs to execute. The goal of Section 4.1 is to evaluate the improvement to LLM reasoning with verifiers (**VALOR-RL vs open-source LLMs; L309)**, and further the impact of verifier-improved visual grounding (**VALOR vs VALOR-RL; L320)**.
> > > >
> > > > In Section 4.4, we report accuracy of VLMs asked to predict the answer directly from (image, query) pairs (**L438-439**). The goal of Section 4.4 is to contrast VLM prediction with our reasoning and tool use paradigm (**L300**).  We again note that we explicitly specify these differences at the introduction of our experiments (**L289-300**), the start of Section 4.1 (**L304-308**), and the start of Section 4.4 (**L437-444)**.
> > > >
> > > > Llama-3.2 achieves two different accuracies in the two tables as in Section 4.1 it is prompted with **only the query** and tasked to write executable Python programs, whereas in Section 4.4 it is prompted with **query and image** and tasked with solving the answer directly. There is no guarantee that these two accuracies should be the same.
> > > >
> > > > We hope our comments have addressed any remaining sources of confusion. If there are further questions or comments we would be happy to address them.

---

### Official Review · Reviewer_DLBC · 2025-10-23

**Soundness:** 4
**Presentation:** 4
**Contribution:** 3
**Rating:** 8
**Confidence:** 4

**Summary:**

The paper addresses the challenging problem of query-based visual reasoning, which requires both accurate object grounding and understanding of complex spatial relationships. The authors propose an annotation-free training framework that integrates two AI-powered verifiers, an LLM verifier that refines reasoning via reinforcement learning, and a VLM verifier that strengthens grounding through hard-negative mining. The approach is evaluated on a variety of spatial reasoning tasks, where it demonstrates superior reasoning and grounding performance compared to both open-source and proprietary baselines.

**Strengths:**

- The paper is well written and clearly structured. It effectively presents both the high-level motivation and the technical components of the method, with helpful examples that aid comprehension.
- It tackles an important and timely problem in visual reasoning, particularly the issue of grounding, which remains a key bottleneck for reliable program-based reasoning systems.
- The annotation-free design is compelling and addresses an important challenge in scaling visual reasoning systems.

**Weaknesses:**

- My only concern lies in the VLM verifier quality. Since the proposed method uses the VLM to generate the training data for grounding, the VLM may itself produce imperfect outputs. Do you think the fine-tuned model can ever outperform the VLM that labeled the data? Did you compare VALOR against GPT-5-mini at some point, as in Table 4? It might be helpful to discuss this topic more explicitly in the paper. For example, should the long-term strategy be to continually use the strongest available VLMs for data generation, or are there other potential directions?

**Questions:**

1. When prompting the LLMs and VLMs, which generation strategy did you use, e.g. greedy generation? For proprietary models, did you use default parameters, or did you adjust reasoning settings (e.g., reasoning effort in GPT-5-mini)?
2. In Table 3, you compare results on only two benchmarks. Is there a specific reason for this selection?
3. In Figure 1, you mention "GPT-5-Thinking", presumably the model option from the OpenAI web app? It might be good to clarify which model these terms refer to (as the API only has gpt-5 and gpt-5-mini, etc.)
4. Line 318: "framing VQA as yes/no rather than open-ended", is this referring to example (c) or to something else? I didn't understand the yes/no framing.
5. I like the various examples given in the paper; however, Figure 4 requires quite some zooming. I think it would improve the paper if the number of examples were reduced or the way the small texts are presented were improved.

---

> ### Author Response · Authors · 2025-11-20
> **Response to DLBC (part 1)**
>
> We thank the reviewer for their thoughtful and thorough evaluation of our work. We are happy you found our paper “well written and clearly structured”, our task “important and timely” and our design “compelling”. We also appreciate your insightful questions and suggestions, which we address below.
>
> > Since the proposed method uses the VLM to generate the training data for grounding, the VLM may itself produce imperfect outputs. Do you think the fine-tuned model can ever outperform the VLM that labeled the data?
>
> This is a great question\! We first remark that we use a VLM to *select* training data for grounding, not *generate* it. We believe this is an important distinction in answering your question: if VLMs are better at verifying detections than they are at generating them, then it is possible for the fine-tuned model to outperform the VLM that *selected* the data.
>
> In fact, we find this to be the case. We use GPT-5-mini as our VLM verifier, which although is highly effective at evaluating object detections, we observe that it often struggles when generating bounding boxes itself. In our updated manuscript, we have included examples in the supplementary (Figure 13; **L932-945**) illustrating this point – GPT-5-mini frequently outputs misaligned or overly large boxes, failing to localize objects that VALOR (trained with GPT-5-mini as a verifier) correctly detects. We believe these examples underscore the important distinction between verification and generation abilities: a VLM can provide reliable binary judgements about correctness even when its own grounding predictions are imperfect.
>
> Moreover, we agree that quantifying the quality of our verifiers is a valuable addition to our work. To provide more context on their performance, we quantify the rate of error of both of our verifiers below.
>
> For our LLM verifier, we sample 100 model outputs during training and manually compute the reward. We consider this set of manually annotated outputs as the ground truth. On the same 100 outputs, we find that our LLM verifier (Gemini-2.5-Flash) matches our ground truth $87\\%$ of the time. We find that on the samples that do not match, Gemini tends to under-reward compared to our ground truth.
>
> Next, for our VLM verifier, we sample 100 outputs from our VLM verification pipeline. For each pipeline stage – coarse filtering, per-crop verification, and deduplication – we recorded:
>
> 1. True positives (TP): correctly identified boxes
> 2. False positives (FP): incorrect boxes or duplicate boxes
> 3. False negatives (FN): missed detections
>
> We compute a per-stage precision metric:
> $$ P \= \\frac{TP}{TP \+ FP \+ FN}$$
>
> We average across all samples, which captures our label precision at each stage. The results are summarized below:
>
> |  | P |
> |--|--|
> | Stage 1 (coarse filter) | 0.450 |
> | Stage 2 (per-crop check) | 0.501 |
> | Stage 3 (de-duplication) | 0.754 |
>
> Our analysis shows that each stage significantly increases label precision. Notably, our final stage of de-duplication yields the largest improvements (+0.253) – this is by design, our VLM verification pipeline hinges on over-prediction by the detector to ensure high recall (**L235)**. This overprediction leads to significant duplicate detections which are subsequently cleaned. We achieve a final label precision of 75% without any human annotations.
>
> We will include both of these analyses in our manuscript.

---

> ### Author Response · Authors · 2025-11-20
> **Response to DLBC (part 2)**
>
> > Did you compare VALOR against GPT-5-mini at some point, as in Table 4?
>
> We report direct prediction accuracy of GPT-5-mini on the same benchmarks below. We note that GPT-5-mini was released after all of the benchmarks, so we cannot rule out data leakage, as it is possible these benchmarks were included in its training corpus. Additionally, to explore how object grounding affects performance for GPT-5-mini, we report an additional variant GPT-5-mini (grounded) where the model is prompted to explicitly predict bounding boxes for the relevant objects in the query prior to a final response.
>
> | | Omni3D-Bench | Robo-Spatial | BLINK | VSR | RealWorld QA | GQA | TallyQA | CountBench QA|
> |--|--|--|--|--|--|--|--|--|
> |GPT-5-mini|40.9|76.3|81.0|84.4|62.2|81.9|55.2|84.3|
> |GPT-5-mini (grounded)|35.5|76.2|76.0|82.4|57.4|80.4|55.4|84.7
> |VALOR|44.0|69.5|69.2|75.6|57.3|64.4|51.0|75.9|
>
> Despite its scale and capabilities, VALOR outperforms GPT-5-mini on Omni3D-Bench, our most reasoning-intensive benchmark. Notably, VALOR – built on a much smaller 8B language-only model – achieves strong results across all benchmarks, which underscores the effectiveness of our verifier-guided training framework.The grounded variant of GPT-5-mini performs similarly to the direct-prediction version on most tasks but degrades on several reasoning-intensive benchmarks, improving only on the counting datasets (TallyQA, CountBench). This suggests that while explicit grounding can help VLMs in object-centric counting scenarios, visually grounded multi-step reasoning without tool-use remains challenging even for large proprietary VLMs.
>
> > Should the long-term strategy be to continually use the strongest available VLMs for data generation, or are there other potential directions?
>
> Thank you for raising this important question, which hinges on the core motivation of our work. We view stronger VLMs/LLMs as increasingly capable verifiers, not merely generators. In fact, we find there are tasks where they are excellent verifiers but not great generators (e.g. object detection, as discussed above). We again make note of the earlier distinction between *selecting* and *verifying* training data as opposed to *generating* it. This suggests an alternative method to improving reasoning in the visual domain – leveraging the multimodal verification capabilities of these models – which is the exact goal of our work. We agree that discussing this topic more explicitly would be helpful to further the motivation of our work, and we are happy to include these ideas in the introduction of our revised manuscript.
>
> > When prompting the LLMs and VLMs, which generation strategy did you use, e.g. greedy generation? For proprietary models, did you use default parameters, or did you adjust reasoning settings (e.g., reasoning effort in GPT-5-mini)?
>
> For inference, we used the recommended parameters specified on the Qwen/Qwen3-8B model card on HuggingFace. Specifically, for the non-thinking variant, these are: Temperature=0.7, TopP=0.8, TopK=20, and MinP=0. During training, we use slightly different parameters to ensure more diverse generations, specified in Table 6 in the Appendix. We used all default parameters for proprietary models, including generation strategy and reasoning effort. We will include all of these parameter details in the paper.

---

> ### Author Response · Authors · 2025-11-20
> **Response to DLBC (part 3)**
>
> > In Table 3, you compare results on only two benchmarks. Is there a specific reason for this selection?
>
> For Table 3 we report on Omni3D-Bench and GQA as those are the benchmarks reported for VisProg, ViperGPT, and VADAR in the VADAR paper. We do this in an attempt to be more favorable to those methods, as they were likely designed for the benchmarks they report.
>
> > In Figure 1, you mention "GPT-5-Thinking", presumably the model option from the OpenAI web app?
>
> That is correct, we used the OpenAI web app for this figure, to have access to the rich multi-modal reasoning trace it produces. We will add this clarification to the text referencing it in the Introduction.
>
> > Line 318: "framing VQA as yes/no rather than open-ended" is this referring to example (c)?
>
> Yes, that line refers to example (c). In this example, the base Qwen3 model asks “Which lane is this car in?” and attempts to parse “opposite” from the answer – this illustrates a failure in the VQA tool use, as it is unlikely the VQA module will respond exactly “opposite”. In contrast, VALOR frames this query as a yes/no question of “Is this car in the opposite lane?” and parses “yes” from the answer. We will revise the text referencing this example for clarity.
>
> > I like the various examples given in the paper; however, Figure 4 requires quite some zooming. I think it would improve the paper if the number of examples were reduced or the way the small texts are presented were improved.
>
> We agree that Figure 4 requires some zooming. Our model produces Python code as output and it is challenging to highlight model behavior without including programs. We have ensured the programs are included as PDFs, so they are legible when zooming in. We attempt to alleviate this by describing the figure contents extensively in the text (**L315-319, 369-373, 408-412**). If you have suggestions for improving the presentation of the programs we would love to incorporate your suggestions.

---

> > ### Comment · Reviewer_DLBC · 2025-11-21
> >
> > Thank you for your detailed response.
> >
> > I apologize for the misunderstanding regarding the selection of train data for the visual grounding task. It makes total sense that judging the quality of bounding boxes is an easier task than generating them.
> >
> > Regarding Figure 4, one option could be to simply outline the general steps of the program, maybe based on the comments included in the program, or summarize the steps in a comprehensible way yourself. By that, the reader understands the general strategy of the program, and it's also easier to understand where the programs differ. The full programs can still be provided, but in my opinion, they would fit better in the appendix.
> >
> > Thanks for the other clarifications as well. My concerns have been addressed, and I maintain my positive assessment of the paper.

---

> > > ### Author Response · Authors · 2025-11-26
> > >
> > > Hi Reviewer DLBC, we are glad we addressed your concerns and thank you for the positive assessment of the paper. We will attempt to incorporate your feedback for the figure into our camera-ready version.

---

### Official Review · Reviewer_3T5W · 2025-10-27

**Soundness:** 3
**Presentation:** 3
**Contribution:** 3
**Rating:** 6
**Confidence:** 5

**Summary:**

This paper introduces VALOR, a verifier-guided, tool-using Visual Language Model (VLM) designed to improve spatial and reasoning-heavy visual understanding tasks without human-annotated labels. The method integrates a language-only model (Qwen3-8B) with structured reinforcement learning through Group Relative Policy Optimization (GRPO) and a multimodal verifier that evaluates generated reasoning programs along six dimensions: logic, spatial, attribute, syntax, adherence, and format. The system further enhances visual grounding by retraining GroundingDINO on verifier-filtered pseudo-labels, improving object detection and spatial relation accuracy. The authors demonstrate strong results across diverse benchmarks such as OMNI3D-BENCH, VSR, and ROBOSPATIAL, outperforming both proprietary and open-source VLMs. The paper provides clean ablations for the reasoning-RL and grounding-improvement stages, showing monotonic performance gains with increasing verifier data. Overall, VALOR presents a unified, label-free approach that aligns structured reasoning, verifier feedback, and improved grounding to achieve significant gains in compositional and spatial visual reasoning.

**Strengths:**

The following are the strenghts of the paper:

**Originality:** The paper presents an integration of verifier-guided reinforcement learning with explicit tool use for visual reasoning, eliminating the need for labeled supervision. The structured multi-head verifier and verifier-filtered pseudo-label pipeline seem to be novel and effective extensions of prior VLM paradigms.

**Technical Quality:** The paper is empirically strong. The experimental design cleanly isolates the effects of reasoning-level RL and grounding-level retraining, supported by consistent scaling studies. The improvements are large and reproducible across benchmarks.

**Clarity:** The paper is clearly written and logically structured. The pipeline is illustrated with detailed figures and algorithmic descriptions that make both the reasoning and grounding stages transparent.

**Significance:** VALOR overcomes a key limitation of current VLMs, weakness in spatial and multi-step reasoning, without extra manual annotation. It generalizes across domains and offers a scalable, verifier-driven way to enhance foundation models, making it practical and influential.

**Weaknesses:**

The following are the weaknesses of the paper:

- The paper introduces a rich multi-head verifier reward but does not ablate the contribution of each component. Since logic, spatial, attribute, syntax, and adherence rewards are argued to fix distinct reasoning errors, omitting a head-wise ablation leaves uncertainty about which rewards are essential versus redundant.

- The three-stage verifier pipeline for generating pseudo-labels in grounding (coarse filter, per-crop verification, deduplication) is central to the claimed improvement, yet there is no quantitative breakdown of each stage’s impact on precision, recall, or downstream accuracy. This weakens the causal link between the pipeline design and the observed performance gains.

- The paper relies on Group Relative Policy Optimization (GRPO) but does not compare it against simpler alternatives such as PPO or supervised fine-tuning on top-rewarded samples. Without this comparison, it is unclear whether the advantage comes from the RL algorithm itself or from the verifier-generated signal.

**Questions:**

The following are the list of questions:

1. Verifier Reward: Can the authors provide an ablation isolating each reward head (logic, spatial, attribute, syntax, adherence) to show which components most affect reasoning performance?

2. Verifier Pipeline: Could the authors quantify how each stage of the pseudo-labeling pipeline (coarse filter, per-crop check, deduplication) impacts label precision and downstream accuracy?

3. RL Baseline: Can the authors compare GRPO with simpler baselines such as PPO or supervised fine-tuning on top-rewarded samples to confirm that GRPO itself drives the improvement?

---

> ### Author Response · Authors · 2025-11-20
> **Response to 3T5W (part 1)**
>
> We thank the reviewer for their detailed and constructive assessment of our work. We appreciate the positive remarks regarding the empirical strength, clarity, and significance of our approach. In particular, we found your suggestions for additional analyses insightful. We address each of these suggestions below, and will incorporate the corresponding analyses into the revised manuscript.
>
> > The paper introduces a rich multi-head verifier reward. Can the authors provide an ablation isolating each reward head
>
> You correctly note that our reward model features various heads that “fix distinct reasoning errors.” We agree that it would provide valuable insight to further analyze the role of each reward head. To this end, we have included a figure in the updated supplementary (Figure 13\) illustrating the role of each reward head. In this figure, we show model outputs during training that failed to achieve a full reward for each of the heads. By focusing on examples that were (correctly) penalized by a particular head, we highlight the types of errors that would otherwise go unaddressed in the absence of that head.
>
> For example, in the Spatial Reward example, the query asks for the “closest plant”, but the model simply takes the first potted plant detected, an omission that would lead to an incorrect answer when multiple plants are present (**L1020-1022**). The Attribute Reward example shows an output where the model does not check the condition of the candles being “on” as required by the query (**L1022-1023**). In the Logic Reward panel, the reward model detects a discrepancy between the VQA prompt in the plan and the original query (**L1023-1024**). Finally, the Code Reward identifies that the generated program does not compare the largest table, even though the plan explicitly says to (**L1025**). In all of these cases we see incorrect model behavior that is **not** rewarded due to the inclusion of the specific heads in our reward model.
>
> We appreciate this suggestion, and hope these additional examples provide more context on the role of each head. We will incorporate this analysis into the manuscript.

---

> ### Author Response · Authors · 2025-11-20
> **Response to 3T5W (part 2)**
>
> > Could the authors quantify how each stage of the pseudo-labeling pipeline (coarse filter, per-crop check, deduplication) impacts label precision
>
> Thank you for the thoughtful suggestion. We agree that quantifying the contribution of each component of the VLM verification pipeline provides valuable insight into where precision is gained and which steps are most critical. To assess this, we conducted a manual inspection of 100 randomly sampled outputs from our VLM verification pipeline. For each pipeline stage – coarse filtering, per-crop verification, and deduplication – we recorded:
>
> 1. True positives (TP): correctly identified boxes
> 2. False positives (FP): incorrect boxes or duplicate boxes
> 3. False negatives (FN): missed detections
>
> We compute a per-stage precision metric:
> $$ P \= \\frac{TP}{TP \+ FP \+ FN}$$
>
> We average across all samples, which captures our label precision at each stage. The results are summarized below:
>
> |  | P |
> |--|--|
> | Stage 1 (coarse filter) | 0.450 |
> | Stage 2 (per-crop check) | 0.501 |
> | Stage 3 (de-duplication) | 0.754 |
>
> Our analysis shows that each stage significantly increases label precision. Notably, our final stage of de-duplication yields the largest improvements (+0.253) – this is by design, our VLM verification pipeline hinges on over-prediction by the detector to ensure high recall (**L235)**. This overprediction leads to significant duplicate detections which are subsequently cleaned. We achieve a final label precision of 75% without any human annotations.
>
> We appreciate your suggestion, and agree that this analysis clarifies the importance of each step in our VLM verification pipeline. We will add this analysis to the manuscript.
>
> > Can the authors compare GRPO with simpler baselines such as PPO or supervised fine-tuning on top-rewarded samples to confirm that GRPO itself drives the improvement?
>
> Although the specific post-training method is not the main focus of our work, we agree that ablating it can shed light on the importance of the nature of the post-training signal. To investigate this, we compare GRPO against supervised fine-tuning, which provides dense supervision over full program outputs rather than the sparse verifier rewards. Following your suggestion, we collect a set of model outputs on the training set with reward $R \\geq 0.7$. We then post-train a Qwen3-8B LLM (same as our base LLM) on these samples using supervised fine-tuning. We evaluate this model equipped with the improved visual grounding module to allow direct comparison to VALOR.
>
> |  | Omni3D-Bench | Robo-Spatial | BLINK | VSR | RealWorld QA | GQA | TallyQA | CountBench QA|
> |--|--|--|--|--|--|--|--|--|
> |SFT|38.3|64.5|67.2|68.2|56.3|60.5|49.5|74.5
> |VALOR|44.0|69.5|69.2|75.6|57.3|64.4|51.0|75.9
>
> We find that SFT consistently underperforms GRPO, particularly on benchmarks where reasoning is the core challenge (e.g., Omni3D-Bench, **L358**), confirming your intuition that GRPO drives the improvement. We will include these findings in the manuscript.

---

> ### Author Response · Authors · 2025-11-25
> **Follow-up response to 3T5W**
>
> Hi Reviewer 3T5W, we thank you again for the time taken to write your review. We wanted to follow up to ask if you had any remaining questions or concerns. Thank you.

---

### Official Review · Reviewer_rpw5 · 2025-11-03

**Soundness:** 3
**Presentation:** 4
**Contribution:** 3
**Rating:** 4
**Confidence:** 3

**Summary:**

The paper proposes a novel visual reasoning training paradigm, VALOR, which enables models to improve both reasoning and visual grounding capabilities without any manual annotations. By allowing the model to tackle visual reasoning tasks through tool invocation, VALOR introduces a two-stage training approach: It first refines the reasoning capability of the base LLM via RL with a Python interpreter and an LLM verifier, which strengthens its ability to plan and generate executable code. Then it fine-tunes the visual grounding module (serving as a tool) using a VLM verifier that generates pseudo-labels. Experiments demonstrate that VALOR surpasses both open-source and proprietary models across a broad range of spatial reasoning benchmarks.

**Strengths:**

- The paper is clearly written and well structured.

- It introduces an innovative training framework that enables the model to jointly improve both itself and the tools it invokes, entirely without human supervision.

- The ablation studies are convincing, showing that verifier-based RL enhances reasoning logic, while verifier-based pseudo-labeling improves visual grounding, with cumulative performance gains when both are combined.

**Weaknesses:**

- The visual grounding module in VALOR is further fine-tuned using verifier-generated pseudo-labels, while all baselines still rely on the frozen pre-trained detector (Table 1). This makes the comparison with baselines partially unfair, since the tools they are allowed to invoke are in fact not properly aligned.

- The paper lacks a quantitative analysis of verifier errors, which would help assess the reliability of verifier supervision.

**Questions:**

- The paper shows that both reasoning and grounding performance increase with the number of training samples (Table 5, Fig. 7).
Have the authors investigated how far this scaling trend continues? Does the performance saturate beyond the current dataset size, or does it continue improving at larger scales?

- The paper mentions using GPT-5-mini as the VLM verifier. Could the authors report its standalone performance on the same benchmarks to better understand the verifier's contribution?

- It would be interesting to explore whether the model could serve as its own verifier for self-training, rather than relying on an external verifier.

- Following Weakness 1, I am interested in understanding the isolated impact of post-training the visual grounding module. Compared to VALOR-RL, this stage appears to contribute the larger performance gain, so disentangling its effect would be insightful.

I would be happy to raise my score if the authors could address these questions and clarify the above points.

---

> ### Author Response · Authors · 2025-11-20
> **Response to rpw5 (part 1)**
>
> We thank you for your thoughtful comments and review. We are glad to hear that you found VALOR to be *“innovative”* and *“novel.”* We address your questions below and will incorporate these additional results into our revised paper.
>
> > The visual grounding module in VALOR is further fine-tuned using verifier-generated pseudo-labels, while all baselines still rely on the frozen pre-trained detector (Table 1\) \[...\] I am interested in understanding the isolated impact of post-training the visual grounding module.
>
> This is a great point. As you note, In Table 1 of the paper, the final VALOR model uses a fine-tuned visual grounding module. However, in the same table, **VALOR-RL** uses the pre-trained detector, ensuring a fair comparison against the baselines.
>
> To additionally showcase the impact of the fine-tuned detector on other models, we report the performance of Gemma-3-12B and Qwen3-8B equipped with **the fine-tuned detector (tuned GD)**. These results are directly comparable to the variants reported in Table 1 – the only difference is the grounding module. We include the baselines with the pre-trained detector (base GD), VALOR-RL (that also uses the base GD) and the full VALOR (that uses the tuned GD) for completeness.
>
> | | Omni3D-Bench | Robo-Spatial | BLINK | VSR | RealWorld QA | GQA | TallyQA | CountBench QA|
> |--|--|--|--|--|--|--|--|--|
> |Gemma3 (base GD)|24.4|54.0|57.4|57.9|47.3|46.0|48.9|67.8|
> |Gemma3 (tuned GD)|25.2|57.0|55.9|60.9|44.7|52.1|49.5|72.7|
> |Qwen3 (base GD)|37.5|60.5|63.9|68.2|53.3|57.4|50.1|68.6|
> |Qwen3 (tuned GD)|37.9|66.7|68.8|72.4|56.2|63.9|50.1|74.1|
> |VALOR-RL (base GD)|43.9|61.8|67.3|70.3|53.5|57.6|49.5|67.6
> |VALOR (tuned GD)|44.0|69.5|69.2|75.6|57.3|64.4|51.0|75.9
>
> We observe that fine-tuning the visual grounding module **improves performance across datasets** for both open-source base LLMs (base vs tuned GD). At the same time, comparing **VALOR** against **Qwen3 with the tuned detector** shows that –  particularly on benchmarks where reasoning is the core challenge (e.g., Omni3D-Bench, **L358**) – improving visual grounding alone is insufficient; the verifier-guided reasoning improvements introduced by our method in §3.1 are important.
>
> We appreciate your suggestion: this ablation isolates contributions from both components – improved grounding and better reasoning – confirming both are needed to achieve the best performance. We will include this analysis in the paper.
>
> > Could the authors report \[GPT-5-mini\] standalone performance on the same benchmarks?
>
> We report direct prediction accuracy of GPT-5-mini on the same benchmarks below. We note that GPT-5-mini was released after all of the benchmarks, so we cannot rule out data leakage, as it is possible these benchmarks were included in its training corpus. Additionally, to explore how object grounding affects performance for GPT-5-mini, we report an additional variant GPT-5-mini (grounded) where the model is prompted to explicitly predict bounding boxes for the relevant objects in the query prior to a final response.
>
> | | Omni3D-Bench | Robo-Spatial | BLINK | VSR | RealWorld QA | GQA | TallyQA | CountBench QA|
> |--|--|--|--|--|--|--|--|--|
> |GPT-5-mini|40.9|76.3|81.0|84.4|62.2|81.9|55.2|84.3|
> |GPT-5-mini (grounded)|35.5|76.2|76.0|82.4|57.4|80.4|55.4|84.7
> |VALOR|44.0|69.5|69.2|75.6|57.3|64.4|51.0|75.9|
>
> Despite its scale and capabilities, VALOR outperforms GPT-5-mini on Omni3D-Bench, our most reasoning-intensive benchmark. Notably, VALOR – built on a much smaller 8B language-only model – achieves strong results across all benchmarks, which underscores the effectiveness of our verifier-guided training framework.The grounded variant of GPT-5-mini performs similarly to the direct-prediction version on most tasks but degrades on several reasoning-intensive benchmarks, improving only on the counting datasets (TallyQA, CountBench). This suggests that while explicit grounding can help VLMs in object-centric counting scenarios, visually grounded multi-step reasoning without tool-use remains challenging even for large proprietary VLMs.

---

> ### Author Response · Authors · 2025-11-20
> **Response to rpw5 (part 2)**
>
> > Quantitative analysis of verifier errors
>
> We agree that a quantitative analysis of verifier errors would be a strong addition to our qualitative analysis (**L897-905**). We provide this analysis below, and will include it in our revised manuscript.
>
> For our LLM verifier, we sample 100 model outputs during training and manually label them. We consider this set of manually annotated outputs as the ground truth. On the same 100 outputs, we find that our LLM verifier (Gemini-2.5-Flash) matches our ground truth $87\\%$ of the time. We find that on the samples that do not match, Gemini tends to under-reward compared to our ground truth.
>
> Next, for our VLM verifier, we sample 100 outputs from our VLM verification pipeline. For each pipeline stage – coarse filtering, per-crop verification, and deduplication – we recorded:
>
> 1. True positives (TP): correctly identified boxes
> 2. False positives (FP): incorrect boxes or duplicate boxes
> 3. False negatives (FN): missed detections
>
> We compute a per-stage precision metric:
> $$ P \= \\frac{TP}{TP \+ FP \+ FN}$$
>
> We average across all samples, which captures our label precision at each stage. The results are summarized below:
>
> |  | P |
> |--|--|
> | Stage 1 (coarse filter) | 0.450 |
> | Stage 2 (per-crop check) | 0.501 |
> | Stage 3 (de-duplication) | 0.754 |
>
> Our analysis shows that each stage significantly increases label precision. Notably, our final stage of de-duplication yields the largest improvements (+0.253) – this is by design, our VLM verification pipeline hinges on over-prediction by the detector to ensure high recall (**L235)**. This overprediction leads to significant duplicate detections which are subsequently cleaned. We achieve a final label precision of 75% without any human annotations.
>
> > It would be interesting to explore whether the model can serve as its own verifier for self-training
>
> This is a fantastic suggestion. Our work proposes a new way to improve spatial reasoning through the use of verifiers – models that do not necessarily have to be good at the task of spatial reasoning, but are good at verifying and providing intermediate rewards. You suggest: can we use this recipe to improve both the base and the verifier using the same model in a dual-purpose fashion? Self-verification is a compelling direction for future work that aligns well with our label-free framework.
>
> Conceptually, VALOR is verifier-agnostic: any model capable of assessing spatial reasoning correctness can serve as the verifier. However, there is some nuance that comes to mind with regard to reward-hacking when the same model is used to generate and evaluate reasoning – namely we must ensure that the model does not trivially assign perfect reward to its own outputs. We attempt an initial experiment with the model serving as its own verifier for self-training. To avoid the issue of collapse, we keep the model frozen when computing reward – ensuring gradients do not incentivize self-rewarding behavior. We include results below, but we note that these are preliminary results and warrant significant further investigation:
>
> | | Omni3D-Bench | Robo-Spatial | BLINK | VSR | RealWorld QA | GQA | TallyQA | CountBench QA|
> |--|--|--|--|--|--|--|--|--|
> |Self-Verifier|36.5|61.8|66.2|74.4|56.2|60.3|50.7|72.7|
> |VALOR|44.0|69.5|69.2|75.6|57.3|64.4|51.0|75.9|
>
> We find the model trained via self-verification trails VALOR across benchmarks. This is not surprising, VALOR uses a stronger, proprietary model as a verifier, while the self-verifier variant uses an 8B model for verification. However, these preliminary results are promising and suggest a potentially interesting direction for self-training spatial reasoning models. We are happy to discuss further if you have any additional suggestions\!
>
> > Does the performance saturate beyond the current dataset size, or does it continue improving at larger scales?
>
> We are eager to explore this question as well. Our scaling analysis (**L456-472**) shows upward trends in performance with data scale. However, pushing beyond our current scale is not feasible within the review period, as both data collection with our verifiers and subsequent model training are time- and compute-intensive.

---

> ### Author Response · Authors · 2025-11-25
> **Follow-up response to rpw5**
>
> Hi Reviewer rpw5, we thank you again for the time taken to write your review. We wanted to follow up to ask if you had any remaining questions or concerns. Thank you.

---

### Meta-Review · Area_Chair_1vwz · 2025-12-25

**Summary:**

The paper introduces a new approach to improve visual spatial reasoning, by training a language-only model with RL and a verifier to evaluate. Multiple reviewers shared similar concerns about whether the numerical results are fair comparisons since the baselines do not have the fine-tuned detector; and whether the verifier errors can be better quantified. The rebuttal includes a series of new ablations answering these questions and demonstrating a fair victory of the proposed method over the baselines under a broad set of scenarios. One reviewer also remarked confusion with the novelty of the method and setup of the experimental design, which the rebuttal attempted to clarify.

I commend the authors on a strong rebuttal clarifying the potential limitations of the method and demonstrating its effectiveness. After adding these results, the paper confirms a strong framework and warrants acceptance. However, I share a similar sentiment with the negative reviewer in requesting better clarity of the novelty of the method and experimental design. I recommend the authors follow the suggestions to improve the clarity of the work in the final version.

**Reviewer Concerns:**

- Whether comparisons with baselines are apples-to-apples since finetuning is performed using verifier-generated pseudo labels versus just the frozen detector in baselines: The rebuttal includes new numerical analysis with a more “fair” comparison showing that existing methods even when equipped with the fine-tuned detector do not outperform the baseline.
- Quantitative analysis of verifier errors and understanding contribution: The rebuttal includes several new analyses of verifier errors to better understand its contributions.
- Experiments with GPT-5 mini: The rebuttal includes additional experiments against GPT-5 mini to demonstrate the method’s advantages.
- Comparisons of GRPO vs PPO or SFT: The rebuttal includes additional experiments against SFT, specifically, to demonstrate the effectiveness of the method.
- Conceptual clarity and experimental design: The rebuttal clarifies many of the questions, but I agree with the reviewer's recommendations to add the clarifying points into the paper.

**Reviewer Scores:**

The rebuttal thoroughly introduces new experiments ablating the effects requested in the reviews. I believe that at least one of the two negative reviewers would have raised their score somewhat. Furthermore, the reviewer with the most negative score continued to engage in discussion with the authors, but mainly presented clarifying questions in the second round, leading me to expect they may increase their score somewhat.

---

### Decision · Program_Chairs · 2026-01-26

Accept (Poster)